# De novo design of protein interactions with learned surface fingerprints

Pablo Gainza[1,2,10,11], Sarah Wehrle[1,2,11], Alexandra Van Hall-Beauvais[1,2,11], Anthony Marchand[1,2,11], Andreas Scheck[1,2,11], Zander Harteveld[1,2], Stephen Buckley[1,2], Dongchun Ni[3,4], Shuguang Tan[5], Freyr Sverrisson[1,2], Casper Goverde[1,2], Priscilla Turelli[6], Charlène Raclot[6], Alexandra Teslenko[7], Martin Pacesa[1,2], Stéphane Rosset[1,2], Sandrine Georgeon[1,2], Jane Marsden[1,2], Aaron Petruzzella[8], Kefang Liu[5], Zepeng Xu[5], Yan Chai[5], Pu Han[5], George F. Gao[5], Elisa Oricchio[8], Beat Fierz[7], Didier Trono[6], Henning Stahlberg[3,4], Michael Bronstein[9 ✉] & Bruno E. Correia[1,2 ✉]

Physical interactions between proteins are essential for most biological processes governing life[1]. However, the molecular determinants of such interactions have been challenging to understand, even as genomic, proteomic and structural data increase. This knowledge gap has been a major obstacle for the comprehensive understanding of cellular protein–protein interaction networks and for the de novo design of protein binders that are crucial for synthetic biology and translational applications[2–9]. Here we use a geometric deep-learning framework operating on protein surfaces that generates fingerprints to describe geometric and chemical features that are critical to drive protein–protein interactions[10]. We hypothesized that these fingerprints capture the key aspects of molecular recognition that represent a new paradigm in the computational design of novel protein interactions. As a proof of principle, we computationally designed several de novo protein binders to engage four protein targets: SARS-CoV-2 spike, PD-1, PD-L1 and CTLA-4. Several designs were experimentally optimized, whereas others were generated purely in silico, reaching nanomolar affinity with structural and mutational characterization showing highly accurate predictions. Overall, our surface-centric approach captures the physical and chemical determinants of molecular recognition, enabling an approach for the de novo design of protein interactions and, more broadly, of artificial proteins with function.

Designing novel protein–protein interactions (PPIs) remains a fundamental challenge in computational protein design, with broad basic and translational applications in biology. The challenge consists of generating amino acid sequences that engage a target site and form a quaternary complex with a given protein. This represents a stringent test of our understanding of the physicochemical determinants that drive biomolecular interactions[11]. Robust computational methods to design de novo PPIs could be used to rapidly engineer protein-based therapeutics such as antibodies and protein inhibitors or vaccines among others, and are therefore of considerable interest for biomedical and translational applications[2–8].

Despite recent advances in rational PPI design[2,6,8] and prediction[12], designing novel protein binders against specific targets is very challenging, particularly when no structural elements from pre-existing binders are known. Current state-of-the-art methods for de novo PPI design[2,6,13,14], such as hotspot-centric approaches[6] and rotamer information fields[2,8], rely on placing disembodied residues on the target interface and then optimizing their presentation on a protein scaffold. Intrinsic limitations of these approaches relate to the very weak energetic signatures provided by scoring functions to single-side chain placements, which is compounded in flat interfaces that lack deep pockets. These methods also face the challenge of finding compatible protein scaffolds to precisely display the generated constellations of residues. To circumvent these limitations, new approaches are needed to design de novo binders to various surface types and protein sites.

A long-standing model of molecular recognition postulates that PPIs form between protein molecular surfaces with chemical and geometric complementarity[15,16]. The complementarity features arise as a

[1]Laboratory of Protein Design and Immunoengineering, Institute of Bioengineering, École Polytechnique Fédérale de Lausanne, Lausanne, Switzerland. [2]Swiss Institute of Bioinformatics, Lausanne, Switzerland. [3]Laboratory of Biological Electron Microscopy, Institute of Physics, School of Basic Science, École Polytechnique Fédérale de Lausanne, Lausanne, Switzerland. [4]Department of Fundamental Microbiology, Faculty of Biology and Medicine, University of Lausanne, Lausanne, Switzerland. [5]CAS Key Laboratory of Pathogen Microbiology and Immunology, Institute of Microbiology, Chinese Academy of Sciences, Beijing, China. [6]Laboratory of Virology and Genetics, School of Life Sciences, École Polytechnique Fédérale de Lausanne, Lausanne, Switzerland. [7]Laboratory of Biophysical Chemistry of Macromolecules, School of Basic Sciences, Institute of Chemical Sciences and Engineering (ISIC), École Polytechnique Fédérale de Lausanne, Lausanne, Switzerland. [8]Swiss Institute for Experimental Cancer Research, School of Life Sciences, École Polytechnique Fédérale de Lausanne, Lausanne, Switzerland. [9]Department of Computer Science, University of Oxford, Oxford, UK. [10]Present address: Monte Rosa Therapeutics, Basel, Switzerland. [11]These authors contributed equally: Pablo Gainza, Sarah Wehrle, Alexandra Van Hall-Beauvais, Anthony Marchand, Andreas Scheck. ✉e-mail: michael.bronstein@cs.ox.ac.uk; bruno.correia@epfl.ch

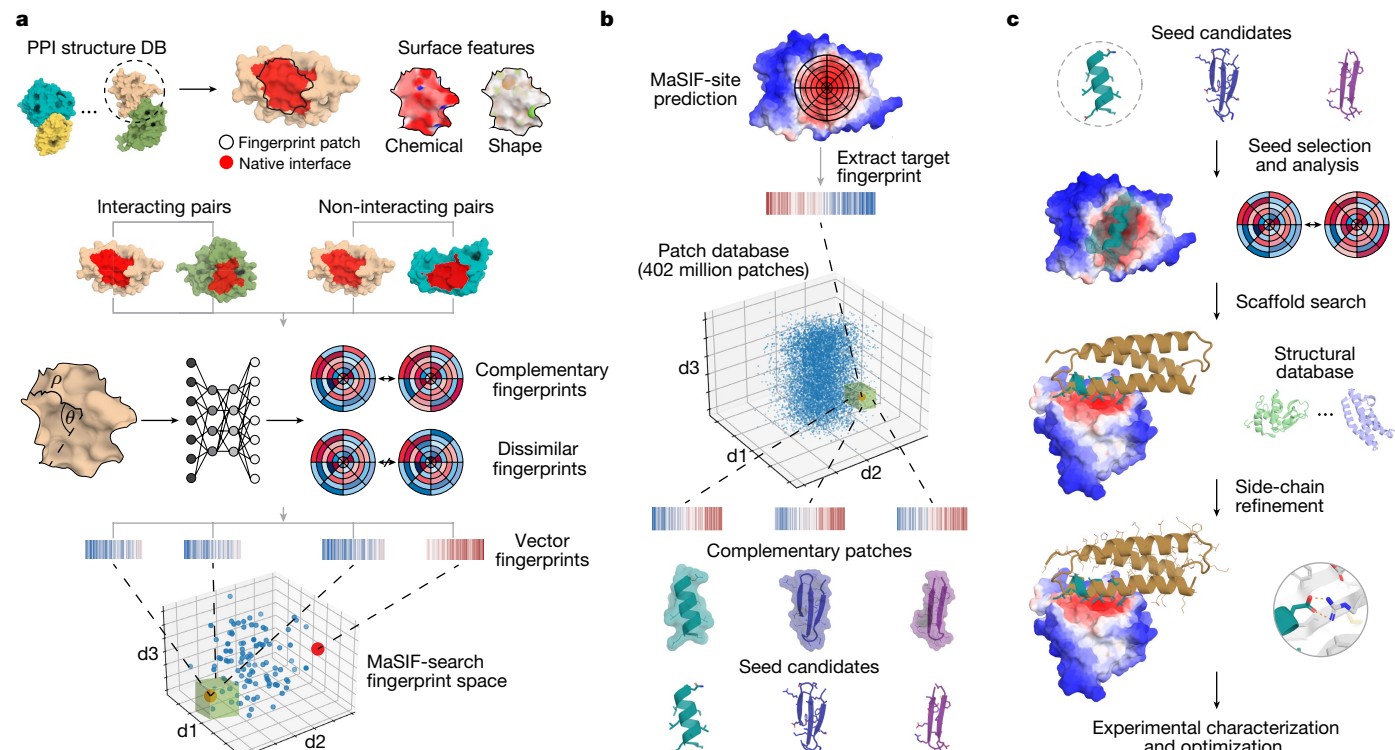

**Fig. 1 | Surface-centric design of de novo site-specific protein binders.** **a**, Schematic of fingerprint generation. Protein binding sites are spatially embedded as vector fingerprints. Protein surfaces are decomposed into overlapping radial patches, and a neural network trained on native interacting protein pairs learns to embed the fingerprints such that complementary fingerprints are placed in a similar region of space. We show an illustration for a subsample of the fingerprints projected in a space reduced to three dimensions. The green box highlights a region of complementary fingerprints. **b**, MaSIF-seed—a method to identify new binding seeds. A target patch is identified by MaSIF-site based on the propensity to form buried interfaces. Using MaSIF-seed, fingerprint complementarity is evaluated between the target patch and all fingerprints in a large database (around 402 million patches); the pairs of fingerprints are subsequently ranked. The top patches are aligned and rescored to enable a more precise evaluation of the seed candidates. **c**, Scaffold search, seed grafting and interface redesign. The selected seeds are transferred to protein scaffolds and the rest of the interface is redesigned using Rosetta. The top designs are selected and tested experimentally.

consequence of the energetic contributions that are critical to stabilize PPIs, including van der Waals interactions (geometric complementarity), hydrophobic effect and electrostatics interactions (chemical complementarity)[15]. At the structural level, most protein interfaces contain surface regions that become inaccessible to solvent after complex formation, which we refer to as buried or the core interface, as well as patches that are involved in the interface but remain solvent-exposed, which we refer to as the interface rim. Residues within the buried areas tend to be much less tolerant to mutations[1,17] and have a large energetic contribution towards the PPI formation, often referred to as hotspots. Rim regions are generally more polar and tolerant to mutations, giving important contributions to affinity and, notably, specificity[1,18]. Guided by these general principles of molecular recognition, we introduce a protein design approach based on the critical importance of the fully buried patches of the interface to drive protein interactions. We implemented these design principles by taking advantage of surface fingerprints learned from interacting protein surfaces that capture features that are determinants for molecular recognition. Our approach enables ultrafast and accurate prediction of privileged sites for PPI design, and reduces the complexity for hotspot search and grafting. We used this design workflow to successfully engineer and characterize binders against four therapeutic targets of interest—namely, SARS-CoV-2 spike, PD-1, PD-L1 and CTLA-4.

## Design strategy and in silico validation

In previous work, we introduced a geometric deep-learning framework—Molecular Surface Interaction Fingerprinting (MaSIF)—to generate surface fingerprints from the geometric and chemical features of molecular surfaces and learn patterns that determine the propensity of protein interactions[10]. Within this framework we developed the MaSIF-site tool to predict areas with a propensity to form PPIs on the surface of proteins. MaSIF-site receives as input a protein decomposed into patches and outputs a per-vertex regression score on the propensity of each surface point to become a buried site within a PPI. We also developed MaSIF-search, another tool to evaluate the surface complementarity between binding partners. MaSIF-search was designed as a Siamese neural network architecture[19] trained to produce similar fingerprints for the target patch versus the binder patch, and dissimilar fingerprints for the target patch versus the random patch. As MaSIF tools had robust performance in PPI-related prediction tasks, we hypothesized that we could use them to design PPIs by targeting sites using only structural information from the target protein. To address the de novo PPI design problem, we devised a three-stage computational approach depicted in Fig. 1: (1) prediction of target buried interface sites with high binding propensity using MaSIF-site (Fig. 1a); (2) surface fingerprint-based search for complementary structural motifs (binding seeds) that display the required features to engage the target site, a protocol we refer to as MaSIF-seed (Fig. 1a,b); (3) binding seed transplantation to protein scaffolds to confer stability and additional contacts on the designed interface (Fig. 1c) using established transplantation techniques[20].

The new MaSIF-seed protocol addresses the problem of identifying binding seeds that can mediate productive binding interactions (Fig. 1 and Extended Data Fig. 1). This task stands as a considerable challenge

## Table 1 | Benchmarking of MaSIF-seed and other docking methods

| | Benchmarked method | No. in top 1 | No. in top 10 | No. in top 100 | No. outside top 100[a] | Average time (min)[b] |
|---|---|---|---|---|---|---|
| Helical seeds | MaSIF-seed | 18 | 18 | 20 | 11 | 15 |
| | ZDock | 3 | 4 | 8 | 23 | 2,715 |
| | ZDock+ZRank2 | 6 | 12 | 21 | 10 | 2,946 |
| Non-helical seeds | MaSIF-seed | 41 | 47 | 49 | 34 | 118 |
| | ZDock | 7 | 9 | 22 | 61 | 2,206 |
| | ZDock+ZRank2 | 21 | 33 | 45 | 38 | 2,400 |

A benchmark of MaSIF-seed against other docking methods in recovering the native binder in the correct conformation from co-crystal structures for 31 helix–receptor complexes or 83 non-helix seed–receptor complexes, discriminating between 1,000 decoys. The number of receptors for which the method recovered the native binding motif (<3 Å iRMSD) within the top 1, top 10 and top 100 results is shown.

[a]The number of receptors for which the method did not recover the native binding motif in the top 100 results.

[b]The average running time, excluding precomputation time.

in protein design owing to the vast space of structural possibilities to explore, as well as the required precision given that subtle atomic-level changes—such as misplaced methyl groups[20,21], uncoordinated water molecules in the interface or incompatible charges—are sufficient to disrupt PPIs[22].

In MaSIF-seed, protein molecular surfaces are decomposed into overlapping radial patches with a 12 Å radius, capturing on average nearly 400 Å$^2$ of surface area, consistent with the buried surface areas observed in native interfaces (Supplementary Fig. 1). For each point within the patch, we compute chemical and geometric features, as well as a local geodesic polar coordinate system to locate points within the patch relative to each other. A neural network is then trained to output vector fingerprint descriptors that are complementary between patches of interacting protein pairs and dissimilar between non-interacting pairs[10] (Fig. 1a and Extended Data Fig. 1). Matched surface patches are aligned to the target site and scored with a second neural network, outputting an interface post-alignment (IPA) score to further improve the discrimination performance of the surface descriptors (Methods).

To benchmark our method, we assembled a test set comprising 114 dimeric complexes, which contained 31 complexes of which the binding motif was a single α-helical segment and 83 of which the binding motif was composed of less than 50% helical segments (Supplementary Fig. 2). As decoy sets, we used 1,000 motifs (ranging from 600,000–700,000 patches), which, in the case of the helical set, also had a helical secondary structure and, in the non-helical set, were composed of two- and three-strand β-sheets.

We benchmarked MaSIF-seed relative to other docking methods to identify the true binder from the co-crystal structure in the correct orientation (interface root mean squared deviation (iRMSD) < 3 Å) among 1,000 decoys (Extended Data Fig. 2). MaSIF-seed identified the correct binding motif in the correct orientation as the top scoring result in 18 out of 31 cases, and 41 out of 83 cases for the helical and non-helical sets, respectively. By contrast, the best performing method, ZDock + ZRank2 (refs. 23–25) identified only 6 out of 31 as top results in the helical set, 21 out of 83 in the non-helical set. In addition to superior performances, MaSIF-seed was considerably faster, showing speed increases of between 20- and 200-fold, which mostly depend on the number of patches derived from each motif. In our benchmark, we also performed comparisons with faster methods, which showed much lower performances than ZDock + ZRank2 (Table 1 and Supplementary Table 1).

An analysis of the cases in which MaSIF-seed performed best showed that its success relied on PPIs of which (1) the interaction site could be correctly identified by the method, and (2) the majority of contacts lie on a radial patch at the interface core, and with a high shape complementarity in that region (Supplementary Fig. 3). This is consistent with how MaSIF-seed was designed to capture protein interfaces using a radial geodesic patch.

Encouraged by MaSIF-seed's speed and accuracy in discriminating the true binders from decoys on the basis of rich surface features, we sought to design de novo protein binders to engage challenging and disease-relevant protein targets. We therefore assembled a motif database including approximately 640,000 structural fragments (402 million surface patches/fingerprints) with distinct secondary structures (approximately 390,000 and 250,000 non-helical and helical motifs, respectively) extracted from the Protein Data Bank (PDB; Methods). We computationally designed and experimentally validated binders against four structurally diverse targets: the receptor-binding domain (RBD) of the SARS-CoV-2 spike protein in which we identified a neutralization-sensitive site; the two partners of the PD-1–PD-L1 complex, an important protein interaction in immuno-oncology that displays a flat interface that is considered to be 'hard to drug' by small molecules (Supplementary Fig. 4); and CTLA-4—another important target for immuno-oncology. We show that our method can be applied to a variety of structural motifs as binding seeds (helical and non-helical), generating functional designs directly from the computational simulations.

## Targeting a predicted SARS-CoV-2 site

We applied our surface-centric approach to design de novo binders to target the SARS-CoV-2 RBD. First, we used MaSIF-site to predict surface sites on the RBD with a high propensity to be engaged by protein binders. We selected a site distinct from the ACE2-binding region, but overlapping such that a putative binder could inhibit the ACE2–RBD interaction (Fig. 2a). At the time, binders to this site were lacking. We searched a subset of our database containing 140 million surface fingerprints derived from helical fragments to find binding seeds that could target the selected site. The 7,713 binding seeds MaSIF-seed provided showed two prominent features: (1) a contact surface devoid of residues with strong binding hotspot features (such as large hydrophobic residues); (2) an equivalent distribution of binding seeds in two distinct orientations of the helical fragment, with the seeds binding at 180° from each other (Fig. 2b), hinting that both binding modes are plausible. Notably, both orientations of the binding seeds present very similar signatures at the surface fingerprint level (Supplementary Fig. 5) and at the sequence level (Fig. 2b).

We synthesized one of the top-ranked binding seeds as a linear peptide, but no binding interaction was detected using surface plasmon resonance (SPR) (Supplementary Fig. 6). Thus, using the Rosetta MotifGraft protocol, we identified several protein scaffolds that were compatible with both binding modes of the seed (Fig. 2c), transplanted the seed hotspot side chains from a top-ranking seed onto the scaffolds and used Rosetta (v.3.13) to optimize the binder interface (Fig. 1c). In total, 63 designs based on 20 scaffolds, with 7–23 mutations relative to the native proteins, were screened in a yeast display analysis (Extended Data Fig. 3a–d). From this initial round of designs, DBR3_01 showed weak binding in yeast display experiments. Moreover, binding of DBR3_01 was competitive with soluble ACE2 (Extended Data Fig. 3e), suggesting that the binder was targeting the correct RBD site. Furthermore, DBR3_01 showed slightly increased binding compared with the native scaffold protein and a variant with a double point mutation at the designed interface residues, further supporting that the seed residues were participating in the binding interaction (Extended Data Fig. 3f and Supplementary Table 2). We next sought to improve the binding affinity of the design by generating two mutagenesis libraries: first, a directed library in the designed interface was prepared (Supplementary Fig. 7),

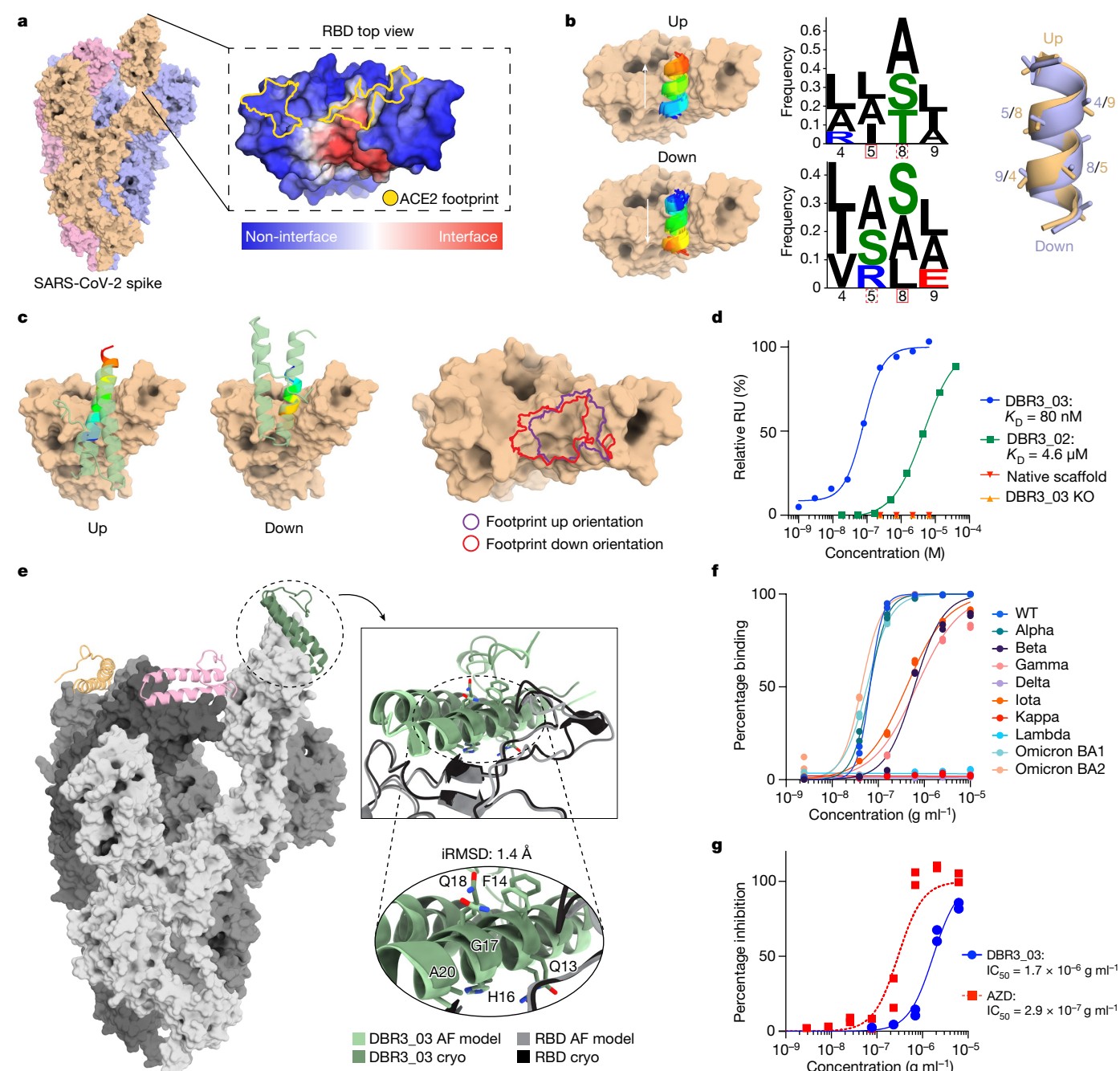

**Fig. 2 | Design and optimization of a SARS-CoV-2 binder targeting the RBD.**
**a**, MaSIF-site prediction of the interface propensity of the RBD. The ACE2-binding footprint (yellow outline) is distinct from the predicted binding site (red). **b**, MaSIF-seed predicts helical seeds that cluster into anti-parallel orientations, referred to as up or down configurations. Sequence logo plots highlight the similarity between the sequences of the two seed clusters, regardless of orientation. **c**, The scaffold (PDB: 5VNY) used to make DBR3_01 allows for binding in the up or down orientation, sharing similar footprints. **d**, SPR data of improved DBR3 binders with controls. DBR3_03 has an affinity of 80 nM with RBD. **e**, A cryo-EM structure (dark green) aligns to the AlphaFold prediction with an iRMSD of 1.4 Å. The trimeric spike protein (grey) has one

DBR3_03 bound per RBD (orange, pink, green). **f**, Fc–DBR3_03 binds to the spike protein of most variants of concern, except for those with the L452R mutation. A list of half-maximal effective concentration ($EC_{50}$) values of DBR3_03 is provided in Supplementary Table 3. The fits were calculated from technical replicates ($n = 2$) using a nonlinear four-parameter curve fitting analysis. **g**, Fc–DBR3_03 neutralizes live Omicron virus in cell-based inhibition assays with an half-maximal inhibitory concentration ($IC_{50}$) of $1.7 \times 10^{-6}$ g ml$^{-1}$, compared with the AstraZeneca (AZD8895 and AZD1061) mix, which has an $IC_{50}$ of $2.9 \times 10^{-7}$ g ml$^{-1}$. The fits were calculated from biological replicates ($n = 2$) using a nonlinear four-parameter curve fitting analysis.

which yielded DBR3_02 with four mutations and a dissociation constant ($K_D$) of 4.6 µM determined by SPR (Fig. 2d and Supplementary Fig. 7). Second, we screened a site-saturation mutagenesis (SSM) library, which resulted in the enrichment of three point mutants, one of which overlapped with a mutation from the first library (Supplementary Fig. 8).

Adding these three mutations to DBR3_02 resulted in DBR3_03, which showed a $K_D$ of 80 nM and was folded and stable (Fig. 2d and Supplementary Fig. 9). Here, we started from a computationally designed binder with very low affinity as observed with yeast display, yet undetectable by SPR, and, after introducing 6 mutations, we observed an improvement

of greater than 60-fold in binding affinity. The mutations all occurred in the binding helix of the design. Of these mutations, A17G and S20A, residing in the core of the interface, appeared to have relieved steric clashes and reduced buried unsatisfied polar atoms, respectively.

To structurally characterize the binding mode of DBR3_03, we solved a cryo-electron microscopy (cryo-EM) structure of the design in a complex with the trimeric spike protein at a local resolution of 2.9 Å (Fig. 2e and Supplementary Figs. 10–12). The structure confirmed the predicted binding sites on both partners. Importantly, the binder adopted the orientation of the helical binding seed that was marginally less favoured by MaSIF's fingerprint descriptors (down orientation) (Fig. 2b). Notably, the initial design DBR3_01 showed similar metrics when the interfaces were analysed in both directions (Supplementary Fig. 5), pointing to known limitations of surface fingerprints in the unbound docking type of problems[10]. This led us to use another state-of-the-art protein docking method, AlphaFold Multimer[26], to predict the complex of DBR3_03 with the spike RBD, and we obtained a 1.4 Å iRMSD between the AlphaFold prediction and the experimental structure (Fig. 2e). This result presents a powerful demonstration of the synergies between machine learning techniques purely based on structural features and those that leverage large sequence-structure datasets for structure prediction tasks. At the structural level, DBR3_03 engages the RBD with a buried interface area of 1,452 Å² (the surface area buried on both sides of the complex), which is much smaller than the average buried surface area of antibodies (approximately $2,071 \pm 456$ Å² (ref. 27), yet still results in a high-affinity interaction. The designed interface lacks canonical hotspot residues and engages the RBD through small residues and is composed of 21% backbone and 79% side-chain contacts. Given the pandemic situation with SARS-CoV-2 and the general need for rational design of protein-based therapeutics to fight viral infections, we next engineered an Fc-fused DBR3_03 (Fc–DBR3_03) construct and tested its neutralization ability on a panel of SARS-CoV-2 variants in virus-free and pseudovirus surrogate assays[28] (Fig. 2f,g, Extended Data Fig. 4a and Supplementary Table 3). We compared the breadth and potency of our design with those of clinically approved monoclonal antibodies. In virus-free assays, we observed that Fc–DBR3_03 had comparable potency to that of imdevimab (REGN10987), an antibody used clinically, for the wild-type (WT) spike and bound to the Omicron strain, whereas RGN87 did not (Extended Data Fig. 4a). Neutralization activity in pseudovirus assays was tested, and Fc–DBR3_03 neutralized Omicron, albeit less potently than the AstraZeneca clinically approved antibody mix (Fig. 2g). A cryo-EM structure showed that the binding mode was nearly identical (1.4 Å backbone RMSD) between the DBR3_03–RBD(WT) complex and the DBR3_03–RBD(Omicron) complex (Supplementary Figs. 13–15). Importantly, Fc–DBR3_03 showed a very broad reactivity to many SARS-CoV-2 variants (Fig. 2f), which is attributable to the sequence conservation of the targeted site and the small binding footprint of the design. The design was sensitive to the L452R/Q mutation present in the Delta, Lambda and Kappa variants (Fig. 2f and Extended Data Fig. 4b), but introducing a single point mutation (L24G) to relieve the clash between L452R and the binder led to the design binding to Delta (Extended Data Fig. 4c). Our results highlight the value of the surface fingerprinting approach to reveal target sites in viral proteins and for the subsequent design of functional antivirals with broad activity.

## Targeting a flat surface in PD-L1

Surface sites presenting flat structural features are difficult to target with small-molecule drugs, leading to their categorization as undruggable. To test our fingerprint-based approach, we sought to design binders to target the PD-1–PD-L1 interaction, which is central to the regulation of T cell activity in the immune system[29]. We used MaSIF-site to find high-propensity protein-binding sites in PD-L1 and, unsurprisingly, the identified site overlapped significantly with the native binding site engaged by PD-1 (Fig. 3a). This site is extremely flat at the structural level,

ranking in the 99th percentile in terms of interface flatness (ranked 7 among 1,068 transient interfaces; Methods and Supplementary Fig. 16), one of the dominant structural features that makes this site hard to drug by small molecules. We next used MaSIF-seed to find binding motifs to engage the site. Among the top results helical motifs clustered in six orientations packing against the β-sheets of PD-L1 (Supplementary Fig. 17). In the most populated cluster (Supplementary Fig. 17), we observed sequence convergence for a 12-residue fragment (Fig. 3b). We next used Rosetta MotifGraft to search for putative scaffolds to display this fragment and used RosettaDesign to optimize contacts at the interface. We tested 16 designs based on 5 different scaffolds for binding to PD-L1 on the surface of yeast. Two designs based on two different scaffolds showed low binding signals (Supplementary Fig. 18), which we refer to as DBL1_01 and DBL2_01 (Fig. 3c). The specificity of the interaction was confirmed by testing hotspot-knockout controls of each design (Supplementary Fig. 18). To improve the binding affinity of DBL1_01, we constructed a combinatorial library with mutations in the predicted binding region, while maintaining the hotspot residues predicted by MaSIF-seed; Extended Data Fig. 5a). From this library, we selected the variant DBL1_02, which has five mutations found mostly in the interface rim of the design that improve the formation of polar contacts. The most substantial change occurred at position 53, a mutation of alanine to glutamine that introduces a hydrogen bond with PD-L1 (Extended Data Fig. 5a). To improve the design's expression and stability, we constructed a second library targeting residues in the protein core to optimize core packing (Extended Data Fig. 5b). Combining mutations from both libraries, we obtained DBL1_03 with 11 mutations from the starting design, which was folded and monomeric in solution, and showed a binding affinity of 2 μM (Fig. 3d and Supplementary Fig. 9), comparable to that of PD-1 ($K_D = 8.2$ μM)[30]. To further assess the optimality of each residue at the interface of the designed binder, we screened a SSM library sampling 19 positions on the basis of DBL1_03. The most relevant positions are shown in Fig. 3f (all positions are shown in Extended Data Fig. 5c,d). The SSM results revealed that the four hotspot residues placed by MaSIF-seed were crucial, as any other residue was deleterious for binding (Fig. 3f). However, in the interface rim, many mutations could provide affinity improvements, strongly suggesting that this region of the interface was suboptimal (Fig. 3f). On the basis of these data, we generated the DBL1_04 variant, which resulted in a tenfold increase in the binding affinity, showing a $K_D$ of 256 nM to PD-L1 (Fig. 3d). Both DBL1_03 and DBL1_04 showed cell-surface binding, comparable to PD-1, on cells expressing PD-L1. The specificity of the designed interaction was confirmed by the binding inability of variants with a single-residue mutation at the interface (Extended Data Fig. 5e).

The second lead design, which uses the same seed but is based on a different scaffold, DBL2_01, could not be solubly expressed and we therefore designed a combinatorial library to improve expression and binding affinity (Extended Data Fig. 5f). From this library, we isolated the variant DBL2_02, which had six mutations, and expressed it in *Escherichia coli*. From the six mutations, three were predicted to be in the interface (Y23K, Q35E, Q42R) and improved binding affinity by forming additional salt bridges with PD-L1 (Extended Data Fig. 5f). The $K_D$ to PD-L1 determined by SPR was 374 nM, more than tenfold higher than the native ligand PD-1. As both designs shared the same binding seed, we transplanted the SSM mutations of the DBL1_04 design and generated DBL2_03, which showed a threefold improvement in binding affinity ($K_D = 120$ nM) (Extended Data Fig. 5i), indicating that the binding seed was engaging PD-L1 in a similar manner to that of DBL1_03. To further assess the influence of each residue in the designed binding interface, we performed an SSM analysis of 19 interface residues of DBL2_03 (Fig. 3f and Extended Data Fig. 5g,h). The SSM profile reiterated that the hotspot residues placed by MaSIF-seed were very restricted in variability, showing that these residues were accurately predicted. By contrast, several positions on the interface rim were suboptimal and mutations to polar amino acids resulted in affinity enhancements.

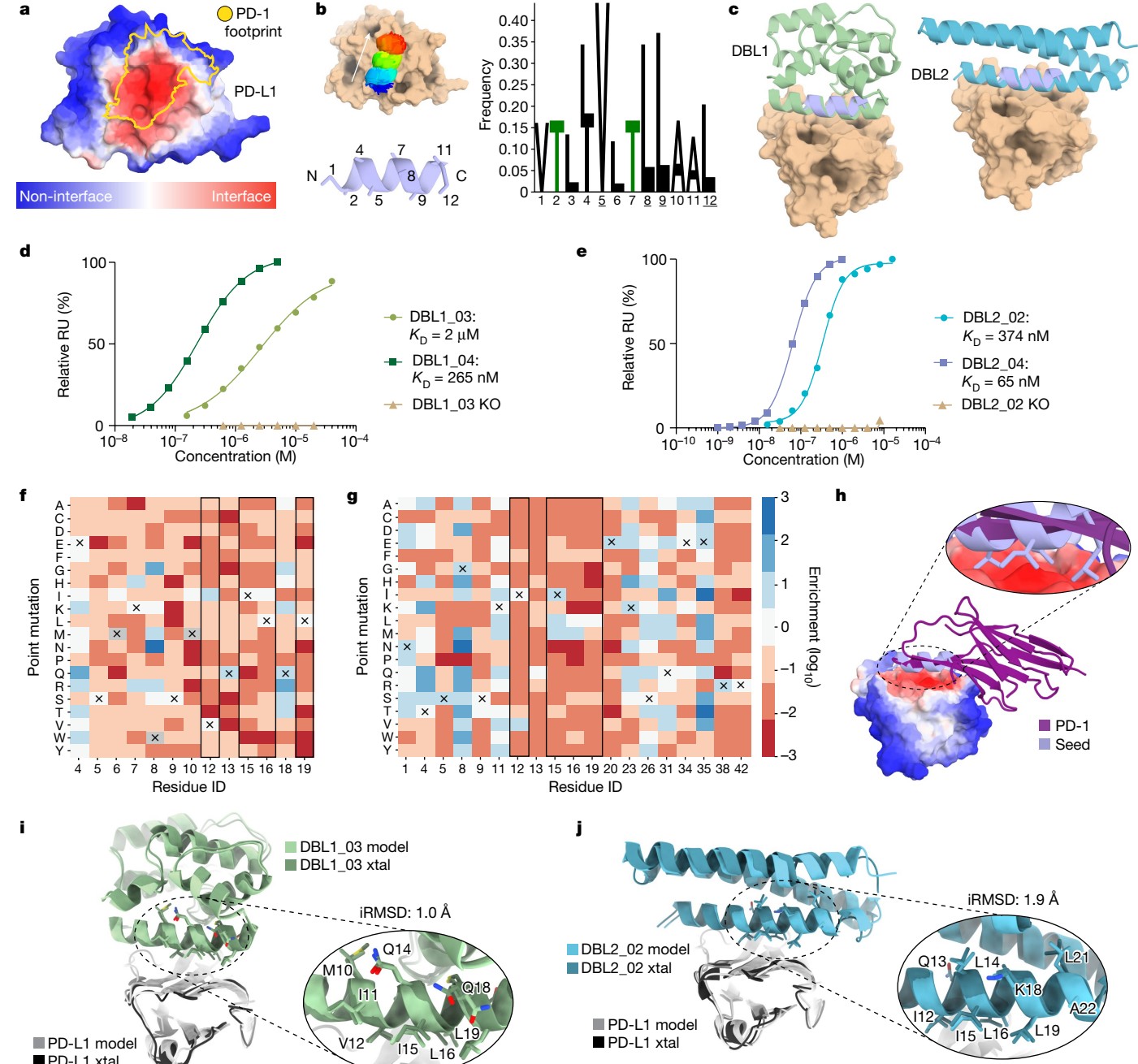

**Fig. 3 | De novo design and optimization of PD-L1 binders targeting a flat surface. a**, MaSIF-site prediction of the interface propensity of PD-L1. The predicted interface (red) overlaps with the binding site of the native interaction partner PD-1 (yellow). **b**, Helical seeds were predicted by MaSIF-seed and clustered. The dominant cluster showed strong amino acid preferences ($Z$-score > 2). Hotspot residues are underlined. **c**, Binders based on two different scaffold proteins using the selected seed were identified. **d**, The binding affinities of DBL1 designs after combinatorial (light green) and SSM library optimization (dark green), measured using SPR. Mutation of a hotspot residue (V12R) ablates binding of DBL1_03 (wheat). **e**, The binding affinities of DBL2 designs after combinatorial (light blue) and SSM library optimization (dark blue), measured using SPR. Mutation of a hotspot residue (V12R) knocks out binding of DBL2_02 (wheat). **f**, SSM analysis of regions of interest in the

binding interface of DBL1_03. The original residue of DBL1_03 is indicated by a cross and hotspot residue positions are shown in black boxes. Enrichment in the binding population (blue) and in the non-binding population (red) is indicated. **g**, SSM data in the binding interface of DBL2_03. The original residue of DBL2_02 is indicated by a cross. **h**, The binding mode of the selected seed in comparison to the native interaction partner PD-1. **i**, Crystal (xtal) structure of DBL1_03 in a complex with PD-L1. The computational model (light green) is aligned with the crystal structure (dark green). Inset: the alignment of the residues in the binding seed. **j**, Crystal structure of DBL2_02 in a complex with PD-L1, shown by aligning the computational model (light blue) with the crystal structure (dark blue). Inset: the alignment of the residues in the binding seed represented as sticks.

On the basis of the SSM data, we generated the DBL2_04 design with additional polar mutations (Fig. 3g and Extended Data Fig. 5i), showing an improved $K_D$ of 65 nM (Fig. 3e). To experimentally validate the binding mode, we co-crystallized the designs with PD-L1 (Supplementary

Fig. 19). Overall, for both designs, the structures (Fig. 3i,j) showed excellent agreement with our computational models, with RMSDs of 0.8 Å and 2.0 Å for the overall backbone and 1.0 Å and 1.9 Å for the full atom interface for DBL1_03 and DBL2_02, respectively, showing a very high

accuracy of the predictions in the interface region. The buried interface area of the designs with PD-L1 was between 1,424 Å² and 1,438 Å², compared with 1,648 Å² for the buried interface area of PD-1 (PDB: 4ZQK). The chemical composition of the designed interface is similar in both designs—around 59% of the surface area is hydrophobic and the remaining area is hydrophilic for DBL1_03 and correspondingly for DBL2_02. These values are comparable to those of the PD-1–PD-L1 interaction (52% hydrophobic surface), showing that we have designed interfaces with chemical compositions similar to the native interaction using a distinct backbone conformation (Fig. 3h). The discovery of binding motifs by MaSIF-seed is notable when comparing the backbone motif used by the native PD-L1-binding partner PD-1 and the designed binders. Whereas the native PD-1 uses a β-hairpin to engage the site, the designed binders do so through an α-helix motif, illustrating the ability of our approach to explore outside of the structural repertoire of native binding motifs. The general trend arising from the designed PD-L1 binders is that, despite the accurate predictions of core residues in the interface, through mutagenesis studies, the designed polar interactions are suboptimal. To address these and other limitations of our computational approach, we performed additional computational design steps to improve the pipeline and tested it on the design of binders to target PD-1.

## One-shot design with native affinities

Despite the successes in designing site-specific binders to engage two different targets, the computational designs still required in vitro evolution to enable expression and detectable binding affinities that could be biochemically characterized. To address these issues, we used a structurally diverse library of binding seeds (helical and β-sheet motifs) and assembled a more comprehensive design pipeline (Fig. 4a), performing: (1) sequence optimization of selected seeds; and (2) biased design for polar contacts in the scaffold interface[31]. To test this approach, we designed de novo binders to target three proteins (PD-L1, PD-1 and CTLA-4). For each of the design targets, we selected the top 2,000 designed sequences according to several structural metrics (Methods) and tested them using yeast display coupled with deep-sequencing readout. According to our deep-sequencing readout, we obtained binders for all three targets using diverse structural motifs to mediate the binding interaction (Supplementary Table 4). Several binders were biochemically characterized to varying degrees. For PD-1, we found three designs based on de novo miniprotein[32,33] scaffolds with interfaces mediated by helical motifs (DBP13_01, DBP40_01 and DBP52_01) (Fig. 4b and Supplementary Fig. 20) that showed a moderate to strong binding signal on the surface of yeast. The most promising candidate binding to PD-1, DBP13_01, was investigated in more detail (Fig. 4b–e). To confirm whether the binding interaction was mediated through the designed interface, we tested several control constructs, which included the native miniprotein scaffold and DBP13_01 variants with predicted knockout mutations (Fig. 4b), all of which abolished binding (Fig. 4c). The interaction site on PD-1 was further probed using a competition assay with nivolumab[34], which blocked the DBP13_01–PD-1 interaction as expected due to the overlapping binding footprints (Extended Data Fig. 6). DBP13_01 did not bind to a close sequence homologue (porcine PD-1), supporting the specificity of the designed interactions (Extended Data Fig. 6). The DBP13_01–PD-1 interaction showed a $K_D$ of 4.2 ± 2 μM ($n$ = 3; Fig. 4d) as determined by SPR, similar to the affinity of the native PD-L1–PD-1 interaction ($K_D$ = 8.2 μM)[30]. This was a promising result given that the design was not subjected to experimental optimization by in vitro evolution. We next performed an SSM experiment and observed that mutations at the predicted core interface positions (Leu23, Leu27, Ile30, Met31) were generally deleterious for binding, supporting the structural and sequence accuracy of the design (Fig. 4e and Supplementary Fig. 21). Moreover, we readily improved the affinity to submicromolar levels by introducing two mutations identified in the SSM data (M31F + H33S, DBP13_02) (Fig. 4d). The predicted complex structure

by AlphaFold Multimer was in agreement with that of MaSIF, with an interface footprint that largely overlaps with the designed residues, and 3.3 Å of backbone RMSD and 2.9 Å of interface full atom RMSD (Supplementary Fig. 22). Although these results are supported by the SSM data, they are a predictive exercise and cannot be interpreted as absolute evidence that the designed binding mode is occurring, which will ultimately require an experimental structure.

Similarly, we experimentally confirmed the specificity of a β-sheet-based-binder to PD-L1 (DBL3_01) (Fig. 4f) with a predicted knockout mutant and a competition assay with high-affinity PD-1 (Fig. 4g and Extended Data Fig. 6). These data were supported by an AlphaFold prediction matching our design model with a 0.97 Å backbone RMSD (Supplementary Fig. 22). Binding to PD-L1 was further improved on yeast by mutating two exposed cysteine residues to serines in the scaffold, which may stabilize the protein and avoid unwanted disulfide bonds (DBL3_02; Fig. 4g and Extended Data Fig. 6). This design adopts a different backbone conformation compared with the native PD-1–PD-L1 interaction, which further demonstrates MaSIF-seed's ability to generalize beyond interactions found in nature (Extended Data Fig. 6). We also estimated the affinity on a yeast display-based assay determining an apparent $K_D$ of 21.8 nM, 42.7-fold higher than the known high-affinity PD-1, which has been reported to have a true $K_D$ of 110 pM (ref. 35) (Fig. 4h).

We also performed experimental characterization for two other binders targeting PD-L1 (DBL4_01) and CTLA-4 (DBC2_01) and observed that the binding interactions are specific to targeted sites by competition and mutagenesis experiments performed using yeast display (Extended Data Figs. 6 and 7). Note that, for several of these binders, the AlphaFold predictions were not in agreement with our models but, nevertheless, the experimental results provide solid evidence that the correct interfaces are involved in the designed interactions (Supplementary Table 4).

Overall, the results show that by starting the interface design process in a manner driven by surface fingerprints and by introducing additional features of native interfaces (such as hotspot optimization and polar contacts), we can design site-specific binders using a variety of structural motifs with native-like affinities purely by computational design.

## Discussion

Physical interactions between proteins in living cells are one of the hallmarks of function[36]. Our incomplete understanding of the complex interplay of molecular forces that drive PPIs has greatly hindered the comprehension of fundamental biological processes as well as the ability to engineer such interactions from first principles. It has been particularly challenging for protein modelling methodologies that use discrete atomic representations to perform de novo design of PPIs[2,6,13,14]. This is in large part due to the small number of molecular interactions that are involved in most protein interfaces and to the very small energetic contributions that determine binding affinities, making physics-based energy functions less reliable[37]. To address this gap, we developed an enhanced data-driven framework to represent proteins as surfaces and learn the geometric and chemical patterns that ultimately determine the propensity of two molecules to interact. We proposed a new geometric deep-learning tool, MaSIF-seed, to overcome the PPI design challenge by both identifying patches with a high propensity to form buried surfaces and binding seeds with complementary surfaces to those patches. By computing fingerprints from protein molecular surfaces, we rapidly and reliably identify complementary surface fragments that can engage a specific target within 402 million candidate surfaces. This, in practice, solves an important challenge in protein design by efficiently handling search spaces of daunting scales.

The identified binding seeds were then used as the interface driving core to design binding proteins against challenging targets: a predicted interface in the SARS-CoV-2 spike protein, which ultimately yielded a

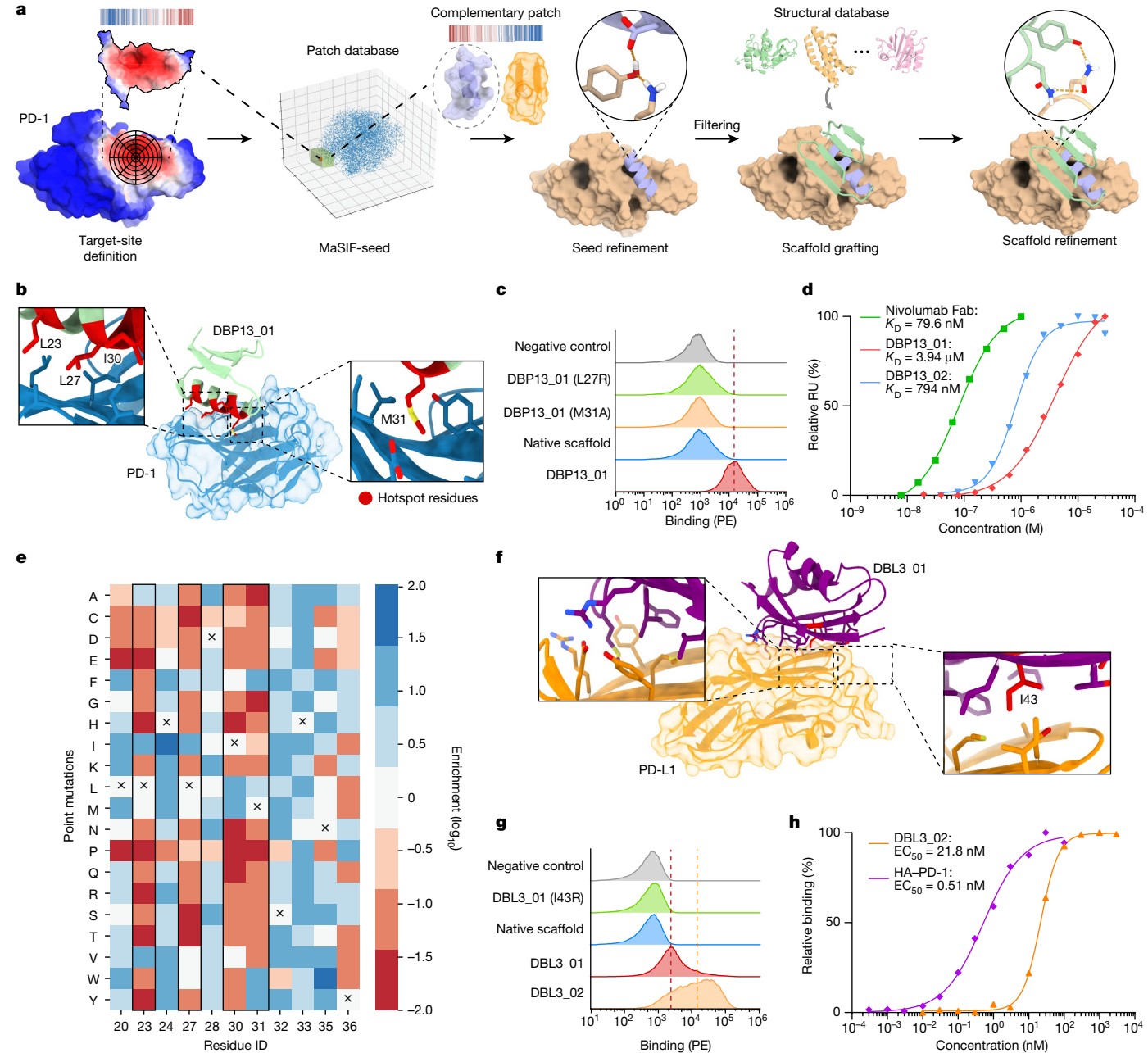

**Fig. 4 | Optimized workflow and de novo binders for PD-1. a**, Improved design computational workflow in which two steps of design are used, at the seed and at the scaffold level, with an emphasis on building new hydrogen bond networks. **b**, PD-1 (blue) targeted by DBP13_01 (green); hotspot residues from the binding seed (red) are highlighted. Insets: crucial residues for binding. **c**, Histogram of the binding signal (PE, phycoerythrin) measured by flow cytometry for DBP13_01, the native miniprotein scaffold, two variants of DBP13_01 with crucial residues mutated and a negative control with unlabelled yeast. The dashed line indicates the geometric mean of the DBP13_01 binding signal. **d**, Binding affinities determined by SPR of the nivolumab Fab (green squares), DBP13_01 (red diamonds) and DBP13_02 (blue triangles). The dissociation constant of DBP13_01 was obtained with three independent measurements. **e**, SSM heat map showing interface residues and the enrichment of each point mutation. The original amino acids in DBP13_01 are indicated by a cross. Enrichment in the binding population (blue) and in the non-binding population (red) is indicated. Hotspot residues are highlighted with a black box. **f**, PD-L1 (orange) targeted by DBL3_01 (purple). Insets: magnification of interface residues, including one crucial residue tested for knockout mutants (Ile43, red). **g**, The binding signal measured using flow cytometry for DBL3_01, DBL3_02, the native protein scaffold, one knockout mutant and a negative control with unlabelled yeast. **h**, PD-L1 ligand titration on yeast displaying DBL3_02 (orange triangle) or high-affinity PD-1 (HA–PD-1, purple diamonds).

SARS-CoV-2 inhibitor, the PD-1–PD-L1 protein complex and CTLA-4, exemplifying sites that are difficult to target with small molecules due to its flat surface. Several designed binders showed close mimicry to computationally predicted models and often achieved high binding affinities after experimental optimization. In the case of purely computationally designed binders, the PD-1 binder showed low micromolar affinity without experimental optimization, which is the range of many native PPIs[38], and several other binders targeting PD-L1 and CTLA-4 were shown to be specific to the targeted sites. Using surface fingerprints, we identified structural motifs that can mediate de novo PPIs, presenting a route to expanding the landscape of motifs that can be used to functionalize proteins and that are critical for the de novo design of function.

For all targets, the original binding seed arguably provided the principal driver of molecular recognition representing the design's binding interface core (Extended Data Fig. 8), maintaining a high surface similarity in this region between the original seed and the final design (Supplementary Fig. 23). However, contacts at the buried interface region are necessary, although, in most cases, probably not sufficient for high-affinity binding. Furthermore, in the three designed binders for PD-L1 and RBD, optimization of the polar interface rim through libraries was necessary to improve binding to a biochemically detectable range ($K_D$ at the micromolar level). Our de novo designs agree with previous findings[6,39] that small changes in the polar interface rim (for example, in the hydrogen bond network surrounding the interface) can result in substantial differences in binding affinities. Encouragingly, by using a larger and more structurally diverse library of binding seeds together with an optimized design pipeline, we obtained several in silico-only designed binders to a variety of targets, representing a major step forward for the robust design of de novo PPIs.

In our study, several limitations of the approach became evident, namely, the absence of conformational flexibility and adaptation of the protein backbone to mutations and the difficulty of designing polar interactions that balance the hydrophobic patches of the interface contributing for affinity and specificity, which has also been observed previously[22,39,40]. In future methodological developments, neural network architectures could be optimized to capture such features of native interfaces. The emergence of generative algorithms that can construct backbones conditioned to the target binding sites or the seed motifs, as recently described by other groups[41,42], presents another exciting route through which our conceptual framework based on surfaces is likely to become more useful to overcome important challenges on the design of molecular recognition.

Here we presented a surface-centric design approach that leveraged molecular representations of protein structures based on learned geometrical and chemical features. We showed that these structural representations can be efficiently used for the design of de novo protein binders—one of the most challenging problems in computational protein design. We anticipate that this conceptual framework for the generation of rich descriptors of molecular surfaces can open possibilities in other important biotechnological fields such as drug design, biosensing or biomaterials in addition to providing a means to study interaction networks in biological processes at the systems levels.

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

# Methods

## Computing buried surface areas

A dataset of PPIs was downloaded from the PDBBind database[43] containing all interactions with a reported affinity stronger than 10 μM; as these PPIs have a reported affinity, all were assumed to be transient. The PDBBind database does not report the chains that are involved in the interaction with the reported affinity; thus, for simplicity, only those complexes containing exactly two chains in the PDB crystal structure were considered for the analysis.

The MSMS program[44] was used to compute all molecular surfaces in this study (density = 3.0, water radius = 1.5 Å). As MSMS produces molecular surfaces with highly irregular meshes, PyMESH (v.0.2.1)[45] was used to further regularize the meshes at a resolution of 1.0 Å. For a given protein subunit that appears in a complex, we define the subunit's buried surface as the patch that becomes inaccessible to water molecules after complex formation. As, in our implementation, a surface is defined by a discretized mesh, we compute the buried surface region as follows. The buried surface of both the subunit and the complex are first independently computed. Then, the minimum distance between every subunit surface vertex and any complex surface vertex is computed. Subunit vertices that are farther than 2.0 Å from a vertex in the surface of the complex are labelled as part of the buried surface as these vertices no longer exist in the surface of the complex. The size of buried areas was determined by computing the area of each vertex labelled as a buried surface vertex.

Note that computing buried surface areas using this method can result in measurements that are different from those widely used in the field, which use the solvent-accessible surface area and count the buried interface of all subunits into a single value (the buried SASA area). Here we use the molecular surface (also known as solvent excluded surface) and count a single subunit. Thus, while in Extended Data Fig. 1 we show areas computed using this method to compare to patch sizes, throughout the rest of this paper we refer to the more widely used buried SASA areas.

## Patch generation in the MaSIF framework

**Decomposing surfaces into radial patches.** To process protein surface information, all molecular surfaces were decomposed into overlapping radial patches. This means that each vertex on the surface becomes the centre of a radial patch of a given radius. To compute the geodesic radius of patches, throughout this Article, we used the Dijkstra algorithm[46]—a fast and simple approximation to the true geodesic distance in the patch. We used a radius size of 12 Å for patches, limited to at most 200 points, which we found corresponds approximately to 400 Å$^2$ (Supplementary Fig. 1), a value close to the median size of the buried interface of transient interactions (Supplementary Fig. 1). Exceptionally, for the MaSIF-site application (described below), we limited the patch to 9 Å or 100 points to reduce the required GPU RAM for this application[10].

**Computing angular and radial coordinates.** An essential geometric deep-learning component in our pipeline is to compute angular and radial coordinates in the patch that enable MaSIF to map features in a 2D plane. The radial coordinate is computed using the Dijkstra algorithm, whereby the geodesic distance (meaning the distance taken to 'walk' along the surface) from the centre of the patch to every vertex is computed. To compute the angular coordinate, all pairwise geodesic distances between vertices in the patch are computed, and the multidimensional scaling algorithm[47] in scikit-learn[48] is then used to map all vertices to the 2D plane. A random direction in the 2D plane is next computed as the 0° frame of reference, and the angle of every vertex in the plane with respect to this frame of reference is computed. Computing the angular and radial coordinates is the slowest step in the MaSIF precomputation. However, we have provided experimental code

to compute these coordinates much faster in our GitHub repository under a branch called 'fast-masif-seed'.

**Geometric and chemical features.** Each point in a patch of the computed molecular surface was assigned an array of two geometric features (shape index[49], distance-dependent curvature[50]), and three chemical features (hydrophobicity[51], Poisson–Boltzmann electrostatics[52] and a hydrogen bond potential[53]). These features are identical to those described previously[10].

**Largest circumscribed patch computation.** From each labelled interface point, we used the Dijkstra algorithm to compute the shortest distance to a non-interface point. The interface point with the greatest distance to a non-interface point was labelled as the centre of the interface, and the distance to the nearest non-interface point as the radius of the largest circumscribed patch.

## Calculation of surface planarity

The surface planarity of all target interfaces with respect to a database of PPIs (Supplementary Fig. 16) was calculated as follows. A total of 690 PPIs crystallized as dimers from the PDBBind database was used as the dataset, resulting in 1,380 interfaces as each chain was analysed separately. Interfaces with an approximate area of lower than 150 Å$^2$ or higher than 1,000 Å$^2$ were discarded, resulting in 1,068 interfaces. The vertices in the buried interface area of each chain were computed, as explained in the 'Computing buried surface areas' section above, and the 3D coordinates of those vertices in the interface were extracted from each chain. The multidimensional scaling method[47] from scikit-learn[48] was then used to position interface vertices in a 2D plane, with the optimization goal of maintaining the distances between all pairs of vertices as close as possible in the 2D embedding as they were in 3D space. The RMSD of all pairwise distances between surface points in the original 3D space versus the 2D space was used as the measure of planarity. Interfaces that are very planar in 3D have small values under this metric as an embedding in 2D preserves the distance between vertices, whereas non-planar interfaces have larger values as an embedding in 2D must significantly alter their 3D distances.

## Geometric deep-learning layer in MaSIF

Geometric deep learning enables the application of traditional techniques from deep learning to data that do not lie in Euclidean spaces, such as a protein molecular surface. At the core of MaSIF lies a mapping from a molecular surface patch to a 2D Euclidean tensor. The mapping is performed through a learned soft polar grid around each patch centre vertex, using the angular and radial coordinates. Once the mapping is performed, a traditional convolutional neural network layer is performed, with an angular max pooling layer, which deals with the rotation ambiguity of geodesic patches. Further details on these techniques were described previously[10,54].

## Prediction of protein interaction sites

The MaSIF-site tool[10] was trained to predict areas with a propensity to form PPIs on the surface of proteins. Here, MaSIF-site was used to predict surface areas with a propensity to form a PPI in 114 targets of our benchmark (Extended Data Fig. 2) and all of the design targets (SARS-CoV-2 RBD, PD-L1, PD-1 and CTLA4). MaSIF-site receives as input a protein decomposed into patches and outputs a per-vertex regression score on the propensity of each point to become a buried surface area within a PPI. MaSIF-site computes a regression score on each point of the surface, yet it becomes necessary to identify the precise patch that we will use to define each interface. Thus, to select interface patches in target proteins, the output of MaSIF-site was decomposed into 12 Å overlapping patches, and the per-vertex prediction for all points in the patch was averaged to obtain a score for each patch.

**Training of MaSIF-site.** MaSIF-site was trained on a database of PPIs sourced from PRISM[55], PDBBind[43], the ZDock benchmark[56] and SabDab[57]. Proteins from these databases that failed to run through the MaSIF pipeline due to, for example, too many incomplete residues in the deposited structure, were discarded. Each instance of these databases, which we refer to as 'subunits' could consist of one or multiple chains (for example, an antibody), and was crystallized in a complex with a partner subunit. In total, 12,002 subunits from deposited structures passed the threshold. These subunits were then clustered by sequence identity at 30% identity and up to one representative from each cluster was selected, resulting in 3,362 subunits. A matrix of all pairwise template modelling (TM) scores for this set was then computed, and a hierarchical clustering algorithm was used on this matrix to split the dataset into 3,004 subunits for the training set and 358 for the testing set.

The molecular surface for each subunit was computed using MSMS[44] and the buried interface area was labelled as described above. The architecture of MaSIF-site (Extended Data Fig. 1b and further described in a previous paper[10]) consisted of three layers of geodesic convolution. The network received as input the full surface of a protein (with batch size of 1) decomposed into overlapping patches of size 9 Å. During training, each vertex of the input was labelled with the ground truth, with a value of 1 if the vertex belonged to the buried area and a label of zero otherwise. The output of the network is a per-vertex assignment of between 0 and 1 for the prediction of that vertex on whether it belongs to the buried surface area or not. A sigmoid activation function was used as the output layer, and a binary cross function as the loss function. Adam[58] was used as the optimization function. MaSIF-site was implemented in Tensorflow (v.1.12)[59], and trained for 40 h on a single-GPU machine, which allowed for 43 epochs. The MaSIF-site neural network implementation in Tensorflow contains a total of 9,267 parameters.

## Complementary surface identification

MaSIF-search[10] was used to compute fingerprints for every overlapping patch in proteins of interest. MaSIF-search was trained on a dataset of 6,001 PPIs (described previously[10]) to receive as input the features of the target, a binder and a random patch from a different protein. MaSIF-search was designed as a Siamese neural network architecture[19] trained to produce similar fingerprints for the target patch versus the binder patch, and dissimilar fingerprints for the target patch versus the random patch. To decrease the training time and improve the performance, the features of the target were multiplied by −1 (with the exception of hydropathy), turning the problem from one of complementarity to one of similarity.

**Training of MaSIF-search.** MaSIF-search was trained on a database of 6,001 PPIs in co-crystal structures sourced from PRISM[55], PDBBind[43], the ZDock benchmark[56] and SabDab[57]. A split between the training and testing set was performed by extracting the atoms at the interface for all 6,001 PPIs and computing a TM-score between all pairs using TM-align. A hierarchical clustering algorithm was used to cluster the pairwise matrix, which was used to split the data into a training set of 4,944 PPIs and a testing set of 957 PPIs. As in MaSIF-site, each side of the interaction could consist of one or multiple chains (for example, an antibody), and we refer to each side as a subunit. In each PPI, pairs of surface vertices within 1.0 Å of each other were selected as interacting pairs.

MaSIF-search produces fingerprints for patches with a radius of up to 12.0 Å in geodesic distance from a central vertex, and is trained to make these patches similar for interacting patches and dissimilar for non-interacting patches (Fig. 1a and Extended Data Fig. 1). We find that MaSIF-search performs best when trained on interacting pairs that lie in the centre of highly complementary interfaces and these pairs were filtered to remove points outside of the interfaces or in interfaces with poor complementarity (described previously[10]).

The MaSIF-search network receives as input the features of a patch from one of these pairs (the binder), the inverted input features of its interacting patch (the target) and a patch randomly chosen from a different interface in the training set (the random patch) (Extended Data Fig. 1). The neural network was trained on a Siamese neural network architecture to produce fingerprints that are similar for the binder and target patches while, at the same time, being dissimilar between target and random. Similarity and dissimilarity were measured as the Euclidean distance between the fingerprints. A total of 85,652 true interacting pair patches and 85,652 non-interacting pair patches was used for training/validation, and 12,678 true interacting and 12,678 non-interacting pairs were used for the testing set.

Each of the five input features was computed in a separate channel consisting of a MaSIF geometric deep-learning convolutional layer. The output from all channels was then concatenated, and a fully connected layer was used to output a fingerprint of size 80. In each batch, 32 pairs of interacting patches and 32 pairs of non-interacting patches were used. Adam was used as the optimizer, and a learning rate of $10^{-3}$ was used. The $d$-prime cost function[60] was used as the loss function. MaSIF-search was trained for 40 h in a GPU, after which it was automatically killed, resulting in 260,000 iterations of the data. The MaSIF-search neural network implementation contained a total of 66,080 trainable parameters and was implemented in Tensorflow.

## Patch alignment and IPA scoring

In the MaSIF-search pipeline, surfaces are computed for each protein of interest, and both a MaSIF-search fingerprint and a MaSIF-site prediction are computed for each surface vertex. All fingerprints within a user-defined threshold for similarity to a target patch (defined at 1.7 by default) are then selected for a second-stage alignment and rescoring. In this step, the patch is extracted from the source protein, along with all the fingerprints for all vertices in the patch (as they were all precomputed). The random sample consensus (RANSAC) algorithm implemented in Open3D[61] then uses the fingerprints of all the vertices in the target and matched patch to find an alignment between the patches. The RANSAC algorithm chooses three random points in the binder patch and computes the Euclidean distance of the surface MaSIF-search fingerprints between these points and all those points in the target patch; the most similar fingerprints provide the RANSAC algorithm with three correspondences to compute a transformation between the patches.

Once a candidate patch is aligned, the IPA neural network is used to score the alignment with a score between 0 and 1 on the prediction of whether the alignment corresponds to a real interaction or not. After patch alignment, each vertex in the candidate patch is matched to the closest vertex in the target patch, and three features are computed per pair of vertices: (1) 1/(distance), the Euclidean distance in 3D between the vertices; (2) the product of the normal between the vertices; and (3) 1/(fingerprint distance), the Euclidean distance between the MaSIF-search fingerprints between the two vertices. A fourth feature, which we call 'penetration' is computed by computing the distance between each of the vertices in the candidate patch and all the atoms in the target. Thus, the IPA neural network receives as input a vector of size $N \times 4$, where $N$ is the number of vertices in the candidate patch (up to 200 vertices). The IPA neural network consists of five layers of one-dimensional convolution, followed by a global averaging pool layer and seven fully connected layers. The five layers of one-dimensional convolution contain 16, 32, 64, 128 and 256 filters, respectively, with a kernel size of 1 and a stride of 1, and each layer was followed by a batch normalization layer and a rectified linear unit layer. The fully connected layers contained 128, 64, 32, 16, 8, 4 and 2 dimensions. Each fully connected layer was also followed by a rectified linear unit layer, with the exception of the last layer, which was followed by a softmax layer. The network was optimized using Adam[58], with a learning rate of $10^{-4}$ and a categorical cross entropy loss function.

The IPA neural network was trained as follows. The same dataset used for MaSIF-search, containing 4,944 PPIs and a testing set of 957 PPIs was used. For each protein pair, one protein was chosen as the target, and the patch at the centre of the interface was selected as the target patch. The partner protein along with ten randomly chosen other proteins were then aligned to it. Any alignment of the true partner within 3 Å RMSD of the co-crystal structure was considered to be a positive. Any alignment from the true partner at greater than that RMSD or of any other protein was considered to be a negative. Features were computed for all alignments and used for the IPA neural network training. The IPA neural network was trained with batches of 32 for 50 epochs.

## Binding seed database
**α-Helix seed library generation.** A snapshot of the non-redundant set of the PDB was downloaded and decomposed into α-helices, removing all non-helical elements. The DSSP program[62] was used to label each residue according to their secondary structure. Fragments with ten or more consecutive residues with a helical (H) label assigned by DSSP were extracted. Each extracted helical fragment was treated as a monomeric protein, and surface features were computed for each one. MaSIF-search fingerprints and MaSIF-site labels were then computed for all extracted helices. MaSIF-seed uses both fingerprint similarity and interface propensity to identify suitable seeds. Ultimately, our binding seed database was composed of approximately 250,000 helical motifs from which 140 million fingerprints were extracted.

**β-Strand seed library generation.** To collect β-strand motifs, a snapshot of the non-redundant set of the PDB was preprocessed with the MASTER software[63] to enable fast structural matches. Two template motifs, one consisting of two β-strands and one consisting of three β-strands, were deprived of loops and served as input to MASTER to find sets of structurally similar motifs that would ultimately become the motif dataset for MaSIF. The search allowed for a variable backbone length of 1–10 amino acids connecting the β-strands of the template. RMSD cut-offs were set at 2.1 Å and 3 Å for two-stranded and three-stranded β-sheets, respectively. Similar to the preparation of helical motifs, each β-fragment was treated as a monomeric protein and surface features were generated, followed by the generation of MaSIF-search fingerprints and MaSIF-site labels. Ultimately, our β-strand binding seed database comprised approximately 390,000 motifs from which 260 million fingerprints were extracted.

## Binding seed identification
On the basis of the different modules within the MaSIF framework[10], we developed a pipeline to identify potential binding seeds to targets. For each target, first MaSIF-seed was used to label each point in the surface for the propensity to form a buried surface region. A fingerprint was then computed for the target site. Finally, after scanning the entire protein, the best patch was selected. In one case, the SARS-CoV-2 RBD, the fourth best site was selected as it was the site with the highest potential to disrupt binding to the natural receptor. A MaSIF-search fingerprint was then computed for the target patch, inverting the target features before inputting them to the MaSIF-search network. The Euclidean distances between the target fingerprint and the millions of fingerprints in the binding seed database were then computed, and all of the patches with a fingerprint distance below the defined thresholds were accepted. In this paper, the thresholds used were <2.0 for PD-L1, PD-1 and CTLA-4, and <1.7 for the RBD.

Once fingerprints are matched, a second-stage alignment and scoring method uses the RANSAC algorithm as described above. After RANSAC produces an alignment, the IPA neural network classifies true binders versus non-binders[10] and outputs an IPA score (described above). Those candidate binders with an IPA score of more than 0.90–0.97 in the neural network score were accepted.

## Computational benchmark
**Helix–receptor motifs.** A set of transient interactions from PDBBind was scanned to identify proteins that bind to helical motifs. A binding motif was determined to be a helix if 80% of residues are helical and the total number of residues does not exceed 60. The selected complexes were filtered to remove pairs of PPIs with high homology and a set of 31 unique PPIs was used; MaSIF-search fingerprints and MaSIF-site fingerprints were subsequently computed. MaSIF-seed was benchmarked against a hybrid pipeline of existing, fast, well-established docking tools on the dataset of helix–receptor proteins: PatchDock[64], ZDock[23,65] and ZRank2 (ref. 24). For each helix–receptor pair, the helix from the co-crystal structure was placed along 1,000 randomly selected helices from the motif database. The methods were then benchmarked to evaluate their ability to rank the correct helix from the co-crystal structure, with an alignment RMSD < 3.0 Å from the conformation of the co-crystal structure, versus the remaining 1,000 helices. Note that each helix can potentially bind in many possible orientations and, in the case of methods that were not preceded by a MaSIF-site prediction of the target site, the helix can bind at many sites on the receptor. The measured time for all methods included only the scoring time, except for MaSIF-seed, for which the alignment time was also included in the calculation.

**MaSIF-seed.** All of MaSIF-seed's neural networks (MaSIF-search, MaSIF-site and the IPA score) were retrained for this benchmark to remove helix–receptor pairs from the training set. In each case, MaSIF-site was used to identify the patch in the target protein with the highest interface propensity, and the fingerprint for the selected patch was compared to the fingerprints of all patches in the database. The rigid orientation of each helix in the benchmark was randomly rotated and translated before any alignment. Patches were discarded if their MaSIF-search fingerprint's Euclidean distance to that of the target site was greater than 1.7. After alignment, patches were further filtered if the IPA score was less than 0.96.

**PatchDock + MaSIF-site.** On each receptor protein, MaSIF-site was used to identify and label the target site, while PatchDock[64] was used to dock all 1,001 helices, setting the target site based on a specific residue using the ReceptorActiveSite flag in PatchDock. The PatchDock score was used to produce the ranking of all conformations for all 1,001 helices.

**ZDock.** ZDock was run on standard parameters and its standard scoring was used similar to PatchDock.

**ZDock + MaSIF-site.** All residues outside of the MaSIF-site-selected patch were blocked using 'compute_blocked_res_list.sh' provided in ZDock.

**ZDock + ZRank2.** In this variant, the top 2,000 results from ZDock with each of the 1,001 peptides for each of the 31 receptors were rescored using ZRank2. The ZRank2 score was then used to score all of the docking poses.

**Non-helix–receptor motifs.** The same set of transient interactions from PDBBind was filtered for proteins interacting through non-helical motifs. The secondary structure types of the proteins were annotated with DSSP[62], followed by computing the contribution of helical segments (DSSP annotation of H, G or I) to the interface. Only interfaces with less than 50% helical segments were selected. Additional filtering was performed by requiring a mean shape complementary at the interface of >0.55 and a maximum inscribed patch area of >150 Å$^2$. From these native complexes, seeds were extracted by selecting residues within a distance of 4 Å to the receptor and extending the backbone of these residues on their N and C terminus until the DSSP annotation changed to capture complete secondary structure elements. In total, 83 complexes were collected for the benchmark.

The decoy set was constructed from 1,000 randomly selected β-strand seeds from the MaSIF-seed pipeline, containing 500

two-stranded and 500 three-stranded β motifs. The benchmark was performed similarly to the helix–receptor benchmark described above with adapting the fingerprint's Euclidean distance cut-off to a value of 2.5 and allowing MaSIF-seed to evaluate the top two sites in each receptor. These modifications were performed for this benchmark as it increased the accuracy while still performing at least 20 times faster than comparable competing tools. Only ZDock and ZDock/ZRank2 were benchmarked in the non-helical benchmark as ZDock/ZRank2 was shown to be the best in the helical benchmark.

## Clustering of seed solutions

In each design case, all of the top matched seeds were clustered by first computing the RMSD between all pairwise helices, computed on the Cα atoms of each pair of helices, in the segment overlapping over the buried surface area. The pairwise distances were then clustered using metric multidimensional scaling[66] implemented in scikit-learn[48].

## Seed and interface refinement

For the one-shot protocol, seed candidates proposed by MaSIF were refined using Rosetta and a FastDesign protocol with a penalty for buried unsatisfied polar atoms in the scoring function[31]. β-Sheet-based seeds containing >33% contact residues found in loop regions were discarded. In total, 33, 200 and 109 refined seeds were selected on the basis of the computed binding energy, shape complementarity, number of hydrogen bonds and counts of buried unsatisfied polar atoms for PD-1, CTLA-4 and PD-L1, respectively.

## Seed grafting and computational design

A representative seed was selected from each solution space, and then matched using Rosetta MotifGraft to a database of 1,300 monomeric scaffolds in the case of the RBD and PD-L1 designs. For the optimized protocol, selected seeds were grafted to a database of 4,347 small globular proteins (<100 amino acids), originating from the PDB[67], two computationally designed miniprotein databases[32,33] and one AF2 proteome prediction database[12,68]. Seeds were cropped to the minimum number of side chains making contact before grafting. Moreover, loop regions from β-sheet-based seeds were completely removed. After side-chain grafting by Rosetta (v.3.13), a computational design protocol was used to design the remaining interface. Final designs were selected for experimental characterization on the basis of the computed Rosetta binding energy, the shape complementarity, the number of hydrogen bonds and counts of buried unsatisfied polar atoms.

## Yeast surface display of single designs

DNA sequences of designs were purchased from Twist Bioscience containing homology overhangs for cloning. DNA was transformed with linearized pCTcon2 (Addgene, 41843) or a modified pNTA vector with V5 tag into EBY-100 yeast using the Frozen-EZ Yeast Transformation II Kit (Zymo Research). Transformed yeast were passaged once in minimal glucose medium (SDCAA) before induction of surface display in minimal galactose medium (SGCAA) overnight at 30 °C. Transformed cells were washed in cold PBS with 0.05–0.1% BSA and incubated with the binding target for 2 h at 4 °C. Cells were washed once and incubated for an additional 30 min with the appropriate antibodies (Supplementary Table 5). Cells were washed and analysed using the Gallios flow cytometer (Beckman Coulter). For quantitative binding measurements, binding was quantified by measuring the fluorescence of a PE-conjugated anti-human Fc antibody (Invitrogen) detecting the Fc-fused protein target. Yeast cells were gated for the displaying population only (V5, MYC or HA positive) (Extended Data Fig. 3a).

## Yeast libraries

Combinatorial sequence libraries were constructed by assembling multiple overlapping primers (Supplementary Table 6) containing degenerate codons at selected positions for combinatorial sampling of the binding interface, core residues or hydrophobic surface residues. Primers were mixed (10 μM each) and assembled in a PCR reaction (55 °C annealing for 30 s, 72 °C extension time for 1 min, 25 cycles). To amplify full-length assembled products, a second PCR reaction was performed, with forward and reverse primers specific for the full-length product. For SSM libraries and oligo pools, DNA was ordered from Twist Biosciences and amplified with primers to give homology to the pCTcon2/pNTA backbone. In all cases, the PCR product was desalted and used for transformation.

## Yeast surface display of libraries

Combinatorial libraries, SSM libraries and oligo pools were transformed as linear DNA fragments at a 5:1 ratio with linearized pCTcon2 or pNTA_V5 vector as described previously into EBY-100 yeast[69]. Transformation efficiency generally yielded around 10^7 transformants per cuvette. Transformed yeast were passaged at least once in minimal glucose medium (SDCAA) before induction of surface display in minimal galactose medium (SGCAA) overnight at 30 °C. Induced cells were labelled in the same manner as the single designs. Labelled cells were washed and sorted using the Sony SH800 cell sorter (acquired using LE-SH800SZFCPL Cell Sorter, v.2.1.5). For combinatorial libraries and oligo pool libraries, sorted cells were grown in SDCAA and prepared similarly for two additional rounds of sorting. After the third sort, cells were plated on SDCAA agar and single colonies were sequenced. SSM libraries were sorted once, collecting both binding and non-binding populations, and grown in liquid culture for plasmid preparation. For flow cytometry analysis of single clones, data were collected with the Galios (Beckman Coulter) cytometer using Kaluza software (Beckman Coulter, v.1.1.20388.18228). Flow cytometry data were analysed using FlowJo (BD Biosciences, v.10.8.1).

## MiSeq Sequencing

After sorting, yeast cells were grown in SDCAA medium, pelleted and plasmid DNA was extracted using the Zymoprep Yeast Plasmid Miniprep II (Zymo Research) according to the manufacturer's instructions. The coding sequence of the designed variants was amplified using vector-specific primer pairs, Illumina sequencing adapters and Nextera barcodes were attached using an additional overhang PCR, and the PCR products were desalted using the Qiaquick PCR purification kit (Qiagen) or AMPure XP selection beads (Beckman Coulter). Next-generation sequencing was performed using the Illumina MiSeq system with appropriate read length, yielding between 0.5–1 million reads per sample. For bioinformatics analysis, sequences were translated in the correct reading frame, and enrichment values were computed for each sequence.

## Protein expression and purification

DNA sequences were ordered from Twist Bioscience and Gibson cloning or T7 ligation was used to clone into bacterial (pET21b) or mammalian (pHLSec) expression vectors. Lists of the protein binder and target constructs are provided in Supplementary Tables 2 and 7, respectively. Mammalian expression was performed using the Expi293 expression system from Thermo Fisher Scientific (A14635). Cells were authenticated and tested negative for mycoplasma contamination (by quantitative PCR) by the provider and no additional authentication and tests were performed. The supernatant was collected 6 days after transfection, filtered and purified. *E. coli* expression was performed using BL21 (DE3) cells and IPTG induction (1 mM at OD 0.6–0.8) and growth overnight at 16–18 °C. Pellets were lysed in lysis buffer (50 mM Tris, pH 7.5, 500 mM NaCl, 5% glycerol, 1 mg ml^{-1} lysozyme, 1 mM PMSF and 1 μg ml^{-1} DNase) with sonication, and the lysate was clarified and purified. All proteins were purified using the ÄKTA pure system (GE healthcare) with either Ni-NTA affinity or protein A affinity columns followed by size-exclusion chromatography. If TEV cleavage was necessary, fused proteins were dialysed overnight at 4 °C (dialysis

buffer: 20 mM Tris pH 7.5, 150 mM NaCl, 10% glycerol) with excess TEV enzymes.

## SPR analysis

SPR measurements were performed on the Biacore 8K (Cytiva; v.4.0.8.19879) system with HBS-EP+ as the running buffer (10 mM HEPES pH 7.4, 150 mM NaCl, 3 mM EDTA, 0.005% (v/v) Surfactant P20, GE Healthcare). Ligands were immobilized on the CM5 chip (GE Healthcare, 29104988) through amine coupling. In total, 500–1,000 response units (RU) were immobilized and designed proteins were injected as an analyte in serial dilutions. The flow rate was 30 µl min$^{-1}$ for a contact time of 120 s followed by 800 s dissociation time. After each injection, the surface was regenerated using 3 M magnesium chloride (for PD-L1) or 10 mM glycine, pH 3.0 (for RBD). Data were fit with a 1:1 Langmuir binding model within the Biacore 8K analysis software (Cytiva, v.4.0.8.19879).

## Biolayer Interferometry

Biolayer Interferometry measurements were performed on the Gator BLI system using the GatorOne software (Gator Bio, v.2.7.3.0728). The running buffer was 150 mM NaCl, 10 mM HEPES pH 7.5. Fc-tagged designs were diluted to 5 µg ml$^{-1}$ and immobilized on the tips (1–2 nm immobilized). The loaded tips were then dipped into serial dilutions of either spike protein or RBD. Curves were fit using a 1:1 model on the Gator software after subtracting the background.

## Size-exclusion chromatography–multi-angle light scattering

Size-exclusion chromatography (controlled by Chromeleon software; Thermo Fisher Scientific, v.7.2.10) with an online multi-angle light scattering device (miniDAWN TREOS, Wyatt) was used to determine the oligomeric state and molecular mass of the protein in solution. Purified proteins were concentrated to 1 mg ml$^{-1}$ in PBS (pH 7.4), and 100 µl of the sample was injected into the Superdex 75 300/10 GL column (GE Healthcare) at a flow rate of 0.5 ml min$^{-1}$, and ultraviolet light (280 nm) and light scattering signals were recorded. Molecular mass was determined using the ASTRA software (Wyatt, v.8.0.2.5).

## Circular dichroism

Far-ultraviolet circular dichroism spectra were measured using a Chirascan spectrometer (AppliedPhotophysics) in a 1 mm path-length cuvette. The protein samples were prepared in a 10 mM sodium phosphate buffer at a protein concentration of between 20 and 50 µM. Wavelengths between 200 nm and 250 nm were recorded with a scanning speed of 20 nm min$^{-1}$ and a response time of 0.125 s. All spectra were averaged twice and corrected for buffer absorption. Temperature ramping melts were performed from 20 to 90 °C with an increment of 2 °C min$^{-1}$. Thermal denaturation curves were plotted by the change of ellipticity at the global curve minimum to calculate the melting temperature ($T_m$).

## Cell binding analysis

Karpas-299 cells were purchased from Sigma-Aldrich (06072604-1VL) with the approval of the European Collection of Authenticated Cell Cultures (ECACC). Cells were authenticated (PCR) and tested negative for mycoplasma contamination (PCR & Vero indicator) by the provider. For flow cytometry analysis of DBL1 designs binding to PD-L1 on Karpas-299 cells, $2 \times 10^5$ cells were incubated with 50 µl Fc Block (BD Biosciences, 553142) that was prediluted 1:50 in FACS buffer (PBS (Gibco/Thermo Fisher Scientific, 10010-015) and 2% BSA (Sigma-Aldrich, A7906)) for 15 min on ice. The samples were subsequently supplemented with 50 µl of PD-L1 binders prepared as follows. High-affinity PD-1_Fc: serially diluted 1:2 for 20 dilutions in FACS buffer, starting at 62.5 µg ml$^{-1}$; DBL1_03_Fc and DBL1_04_Fc: serially diluted 1:2 for 16 dilutions in FACS buffer, starting at 125 µg ml$^{-1}$; DBL1_03_KO_Fc and PD-1_Fc: serially diluted 1:2 for 14 dilutions in FACS buffer, starting at 125 µg ml$^{-1}$.

The cell solutions were incubated for 30 min. The samples were then washed three times, resuspended in 100 µL of FACS buffer containing secondary R-PE goat anti-human IgG antibody diluted 1:100 (Jackson ImmunoResearch, 109-117-008) and incubated for 30 min. The samples were then washed three times to remove unbound antibody, resuspended in 100 µl of FACS buffer, and analysed using the LSR Fortessa flow cytometer (BD Biosciences).

## Protein purification for crystallography

PD-L1 extracellular domain fragment (UniProt: Q9NZQ7; from Phe19 to Arg238) was overexpressed as inclusion bodies in the BL21 (DE3) strain of *E. coli*. Renaturation and purification of PD-L1 was performed as previously described[70]. In brief, inclusion bodies of PD-L1 were diluted against a refolding buffer (100 mM Tris, pH 8.0, 400 mM L-arginine, 5 mM EDTA-Na, 5 mM glutathione (GSH), 0.5 mM glutathione disulfide (GSSG)) at 4 °C for 24 h. The PD-L1 was then concentrated and exchanged into a buffer of 20 mM Tris-HCl (pH 8.0) and 15 mM NaCl and further analysed using HiLoad 16/60 Superdex 75 pg (Cytiva) chromatography. PD-L1 binder designs (DBL1_03 and DBL2_02) were overexpressed in *E. coli* as inclusion bodies. Renaturation and purification of the PD-L1 binder designs were performed as for the PD-L1 protein. PD-L1 and binder designs were then mixed together at a molar ratio of 1:2 and incubated for 1 h on ice. The binder–PD-L1 complex was further purified by HiLoad 16/60 Superdex 75 pg (Cytiva) chromatography.

## Data collection and structure determination

For crystal screening, 1 µl of binder–PD-L1 complex protein solution (10 mg ml$^{-1}$) was mixed with 1 µl of crystal growing reservoir solution. The resulting mixture was sealed and equilibrated against 100 µl of reservoir solution at 4 or 18 °C. Crystals of the DBL1_03–PD-L1 complex were grown in 0.2 M potassium formate and 20% (w/v) PEG 3350. Crystals of the DBL2_02–PD-L1 complex were grown in 0.2 M potassium/sodium tartrate, 0.1 M Bis-Tris propane, pH 6.5 and 20% (w/v) PEG 3350. Crystals were flash-cooled in liquid nitrogen after incubating in anti-freezing buffer (reservoir solution containing 20% (v/v) glycerol). Diffraction data of crystals were collected at Shanghai Synchrotron Radiation Facility (SSRF) BL19U. The collected intensities were subsequently processed and scaled using the XDS package[71] (v.Jan 10 2022, BUILT = 20220220). The structures were determined using molecular replacement with the program Phaser MR in PHENIX (v.1.20.1-4487), with the reported PD-L1 structure (PDB: 3RRQ) as the search model[72]. COOT (v.0.9.5) and PHENIX (v.1.20.1-4487) were used for subsequent model building and refinement[73,74]. The stereochemical qualities of the final model were assessed using MolProbity[75] (v.4.5.1). Details of data collection and refinement statistics are shown in Extended Data Table 1.

## Luminex binding assays

Luminex beads were prepared as previously published[28]. In brief, MagPlex beads were covalently coupled to SARS-CoV-2 spike proteins of different variants. The serial dilutions of the antibodies or design were performed and binding curves were fit using Prism (GraphPad, v.9) nonlinear four-parameter curve fitting analysis of the log[agonist] versus response.

## Live virus neutralization assays

The virus neutralization assays were performed as previously published[28]. In brief, VeroE6 cells were seeded into 96-well plates the day before the infection. The DBR3_03-Fc compound in serial dilutions was mixed with Omicron-spike virus and incubated at 37 °C for 1 h before addition to the cells. The cells with virus were kept a further 48 h at 37 °C, and then washed and fixed for crystal violet staining and analysis. Neutralization EC$_{50}$ calculations were performed using Prism (GraphPad, v.9) nonlinear four-parameter curve fitting analysis.

## Cryo-EM preparation and data acquisition

For cryo-electron microscopy investigations, 3.0 µl aliquots at a concentration of 0.87 or 1.0 mg ml$^{-1}$ of the spike(D614G)–binder sample or the spike(Omicron)–binder sample were applied onto glow-discharged carbon-coated copper grids (Quantifoil R2/1,400 mesh), blotted for 4.0–8.0 s and flash-frozen in a liquid ethane/propane mixture cooled to liquid nitrogen temperature, using the Vitrobot Mark IV (Thermo Fisher Scientific) with 100% humidity and the sample chamber operated at 4 °C. The grids were screened in the Thermo Fisher Scientific 200 kV Glacios cryo-EM instrument. Suitable grids were transferred to TFS Titan Krios instruments for data collection. Cryo-EM data collection statistics of this study are summarized in Extended Data Table 2. The spike(D614G)–binder data comprising 20,794 videos were collected on the Titan Krios G4 microscope, equipped with a cold-FEG electron source and operated at 300 kV acceleration voltage. Videos were recorded with the automation program EPU (Thermo Fisher Scientific, v.2.12.1) on a Falcon4 direct electron detector in counting mode at a physical pixel size of 0.40 Å per pixel and a defocus ranging from −0.8 to −2.0 µm. Exposures were collected as electron event recordings (EER) with a total dose of 80 e$^-$ Å$^{-2}$ over approximately 3 s, corresponding to a dose rate of 4.53 e$^-$ px$^{-1}$ s$^{-1}$. For spike(Omicron)–binder data, 22,266 videos were recorded on the Titan Krios G4 microscope, equipped with TFS SelectrisX imaging filter and Falcon4 camera. Exposures were collected at 60 e$^-$ Å$^{-2}$ total dose with a physical pixel size of 0.726 Å per pixel over approximately 6 s, corresponding to a dose rate of 5.4 e$^-$ px$^{-1}$ s$^{-1}$, at a defocus range of −0.8 to −2.5 µm. Data were analysed using cryoSPARC (v.3.3.1)[76].

## Cryo-EM image processing

Details of the image processing are shown in Supplementary Figs. 10–15 and Extended Data Table 2. Recorded videos in EER format were imported into cryoSPARC (v.3.3.1)[76] and gain-normalized, motion-corrected and dose-weighted using the cryoSPARC implementation of patch-based motion correction. CTF estimation was performed using the patch-based option in cryoSPARC. A small set of particles was manually selected and followed by 2D classification to create a 2D template for the subsequent automatic particle picking. For the sample of spike(D614G) in complex with the de novo designed binder, 832,816 particles were automatically selected by a template-based picker and processed for three rounds of 2D classification, resulting in a particle set of 184,763 particles. The particles were grouped into three classes, using the ab initio and hetero-refine implementations in cryoSPARC. The best 3D class, comprising 97,804 particles, was further processed for another round of ab initio reconstruction and hetero-refinement. The well-resolved class consisting of 67,432 particles resulted in a 2.6 Å overall resolution global map in $C_1$ symmetry. The binder–RBD region was refined with a soft mask, resulting in a local map at 3.1 Å resolution. For the data processing of the spike(Omicron)–binder complex sample, 1,820,333 particles were picked using the cryoSPARC template-based picker. After two rounds of 2D classifications, 981,561 particles were selected and processed for ab initio reconstruction and hetero-refinement, resulting in a set of 595,599 particles. Subsequently, the selected particle set was classified by multiple rounds of 3D classifications in cryoSPARC. The best-resolved 3D class, containing 50,758 particles, resulted in a 2.8 Å overall resolution map and the binder–RBD region was further improved by performing focused refinement with a soft mask, resulting in a map at a resolution of 3.3 Å. The resolution for all 3D maps was estimated based on the Fourier shell correlation with a cut-off value of 0.143.

For model building of the spike(D614G)–binder, the previous model (PDB: 7BNO; spike(D614G)) was used for the region of spike(D614G) as a starting model. The model was rigid-body fit into the cryo-EM density in UCSF Chimera[77] and adjusted manually in Coot (v.0.9.4)[78].

De novo building for the binder parts was performed manually in Coot (v.0.9.4). For building the spike(Omicron)–binder structure, the model (PDB: 7QO7, spike(Omicron)) was fitted into the density and rebuilt and adjusted manually using UCSF Chimera and Coot (v.0.9.4). After the structural rebuilding, all of the atomic models were refined using the Phenix (v.1.19.2-4158) implementation of real.space.refine with general structural restraints[79,80]. Comprehensive validation (cryo-EM), model quality assessment and statistics are provided in Extended Data Table 2. EM densities and atomic models were visualized in ChimeraX (UCSF, v.1.3)[81] and Pymol (Schrödinger, v.2.0).

## Reporting summary

Further information on research design is available in the Nature Portfolio Reporting Summary linked to this article.

## Data availability

Cryo-EM maps were deposited in the Electron Microscopy Data Bank under the following accession codes: EMD-14947 (spike(D614G)–binder full and spike(D614G)–binder local maps), EMD-14922 (spike(Omicron)–binder full) and EMD-14930 (spike(Omicron)–binder local). Atomic models were deposited at the PDB under the following accession codes: 7ZSS (spike(D614G)–binder), 7ZRV (spike(Omicron)–binder full) and 7ZSD (spike(Omicron)–binder local). Crystal structures have been deposited at the PDB under the following accession codes: 7XYQ (DBL1_03–PD-L1 complex) and 7XAD (DBL2_02–PD-L1 complex). The PDBbind database (2018 release), PRISM database, ZDock benchmark and SabDab database, respectively are available online (http://pdbbind.org.cn/index.php; http://cosbi.ku.edu.tr/prism; https://zlab.umassmed.edu/benchmark/; http://opig.stats.ox.ac.uk/webapps/sabdab). The scaffold database generated for grafting the seed provided by the MaSIF-seed is available at Zenodo (https://zenodo.org/record/7643697#.Y-z533ZKhaQ).

## Code availability

MaSIF-seed and the Rosetta design scripts are available at GitHub (https://github.com/LPDI-EPFL/masif_seed).

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

**Acknowledgements** We thank the staff at the Dubochet Center for Imaging (DCI) in Lausanne for cryo-EM data collection. The DCI is an initiative of the EPFL, University of Lausanne and University of Geneva. D.N. and H.S. were supported by the Swiss National Science Foundation and the NCCR Transcure; M.B. by an ERC Consolidator grant no. 724228; B.E.C. by the Swiss National Science Foundation, the NCCR in Chemical Biology, the NCCR in Molecular Systems Engineering and the ERC Starting grant no. 716058. P.G. was sponsored by an EPFL-Fellows grant funded by an H2020 Marie Skłodowska–Curie. We thank K. Lau and F. Pojer from the PTPSP facility at EPFL for providing SARS-CoV-2 spike proteins and assistance with cryo-EM; the staff at SCITAS at EPFL for support in the computational simulations; the staff at GECF for assistance with deep sequencing; members of FCCS for assistance in FACS; and E. Levy, S. Fleishman and M. Azoitei for their feedback on the manuscript.

**Author contributions** Z.H., S.B. and D.N. contributed equally to this work. P.G., S.W., A.V.H.-B., A.M., A.S., M.B. and B.E.C. conceived the work and designed the experiments. P.G., A.M., A.S. and Z.H. performed the computational design. S.W., A.V.H.-B., S.B. and A.M. performed experimental characterization and optimization. P.G., F.S., A.M., Z.H. and A.S. developed the MaSIF-seed method. D.N. and H.S. solved the cryo-EM structure. S.T., M.P., K.L., Z.X., Y.C., P.H. and G.F.G. solved the crystal structures. A.P. and E.O. performed the PD-L1 cell binding assay. A.T. and B.F. synthesized peptides. P.T., C.R. and D.T. performed SARS-CoV-2 binding and neutralization studies. F.S., C.G., S.R., S.G. and J.M. performed experiments and acquired data. P.G., S.W., A.V.H.-B., A.M., A.S. and B.E.C. wrote the manuscript with input from all of the authors.

**Funding** Open access funding provided by EPFL Lausanne.

**Competing interests** Ecole Polytechnique Fédérale de Lausanne (EPFL) has filed a provisional patent application that incorporates findings presented in this Article. P.G., S.W., A.V.H.-B., A.M., A.S., Z.H., F.S., M.B. and B.E.C. are named as co-inventors on this patent (European Patent Office, EP22177692.5).

**Additional information**
**Correspondence and requests for materials** should be addressed to Michael Bronstein or Bruno E. Correia.

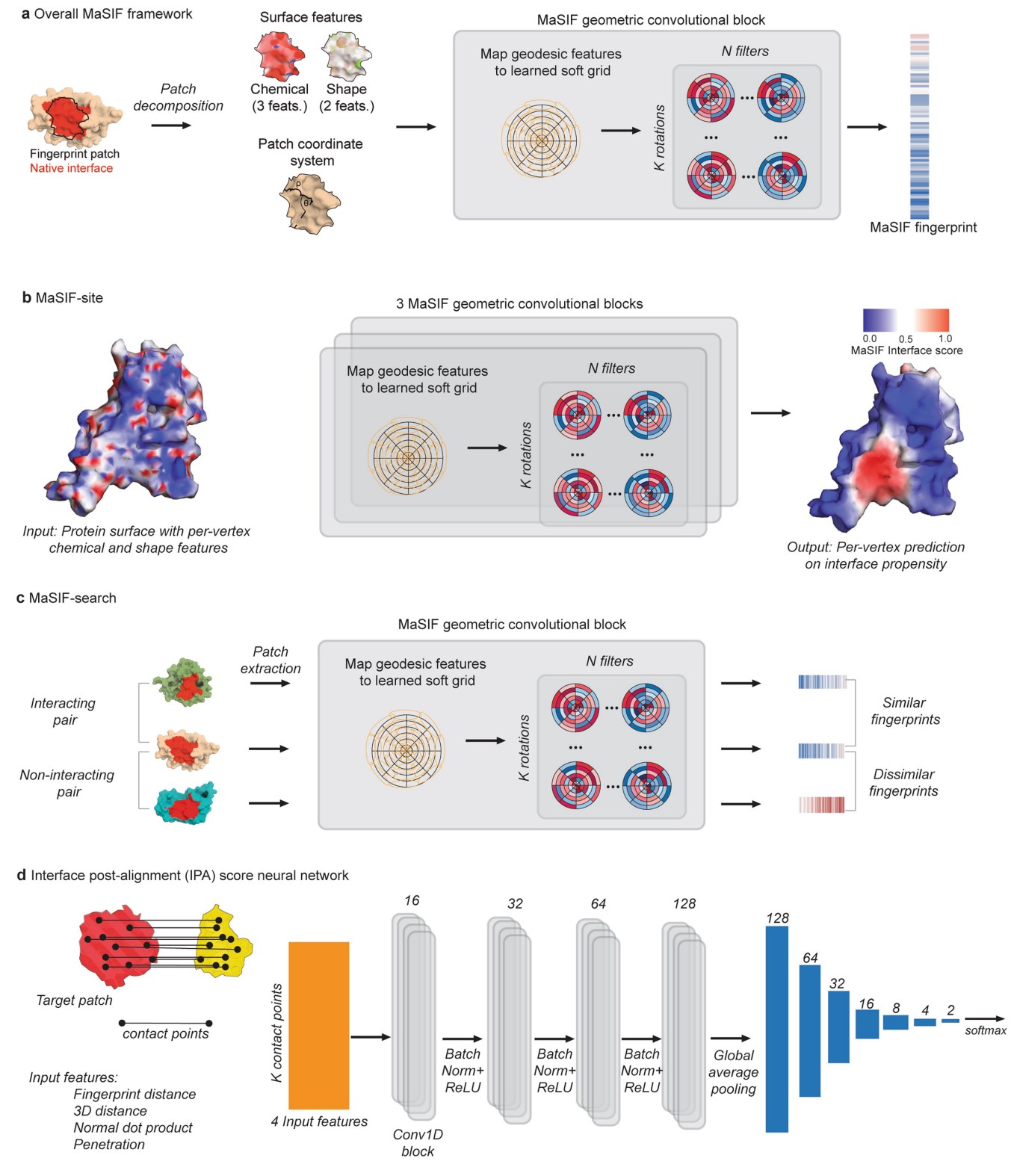

**a** Overall MaSIF framework

Surface features

Chemical (3 feats.)   Shape (2 feats.)

Patch decomposition

Fingerprint patch
Native interface

Patch coordinate system

MaSIF geometric convolutional block

Map geodesic features to learned soft grid

N filters

K rotations

MaSIF fingerprint

**b** MaSIF-site

3 MaSIF geometric convolutional blocks

Map geodesic features to learned soft grid

N filters

K rotations

0.0   0.5   1.0
MaSIF Interface score

Input: Protein surface with per-vertex chemical and shape features

Output: Per-vertex prediction on interface propensity

**c** MaSIF-search

Interacting pair

Non-interacting pair

Patch extraction

MaSIF geometric convolutional block

Map geodesic features to learned soft grid

N filters

K rotations

Similar fingerprints

Dissimilar fingerprints

**d** Interface post-alignment (IPA) score neural network

Target patch

contact points

Input features:
Fingerprint distance
3D distance
Normal dot product
Penetration

K contact points

4 Input features

16   32   64   128

Conv1D block

Batch Norm+ ReLU   Batch Norm+ ReLU   Batch Norm+ ReLU

Global average pooling

128   64   32   16   8   4   2

softmax

**Extended Data Fig. 1 | Overview of the neural network architectures used in the MaSIF protocols. a**, General MaSIF framework. Molecular surfaces are decomposed into patches which are annotated with chemical and shape features. The MaSIF network translates these input features into fingerprints that describe the original surface patch. **b**, MaSIF-site neural network. MaSIF-site predicts partner-independent protein interface propensities based on per-vertex chemical and shape features of the protein surface. **c**, MaSIF-search neural network. MaSIF-search embeds protein patches into a space where complementary patches are close to each other. The network was trained on discriminating interacting patches from non-interacting protein surface patches. The network uses MaSIF fingerprints to identify which are compatible and therefore to predict likely interacting proteins. **d**, Interface post-alignment (IPA) scoring neural network. The IPA scoring neural network enables the scoring of protein interfaces based on several input features: fingerprint distance between contacting points, 3D distance of corresponding points, normal dot product, and the distance between surface points in the seed and the closest atom in the target, which we call 'penetration'.

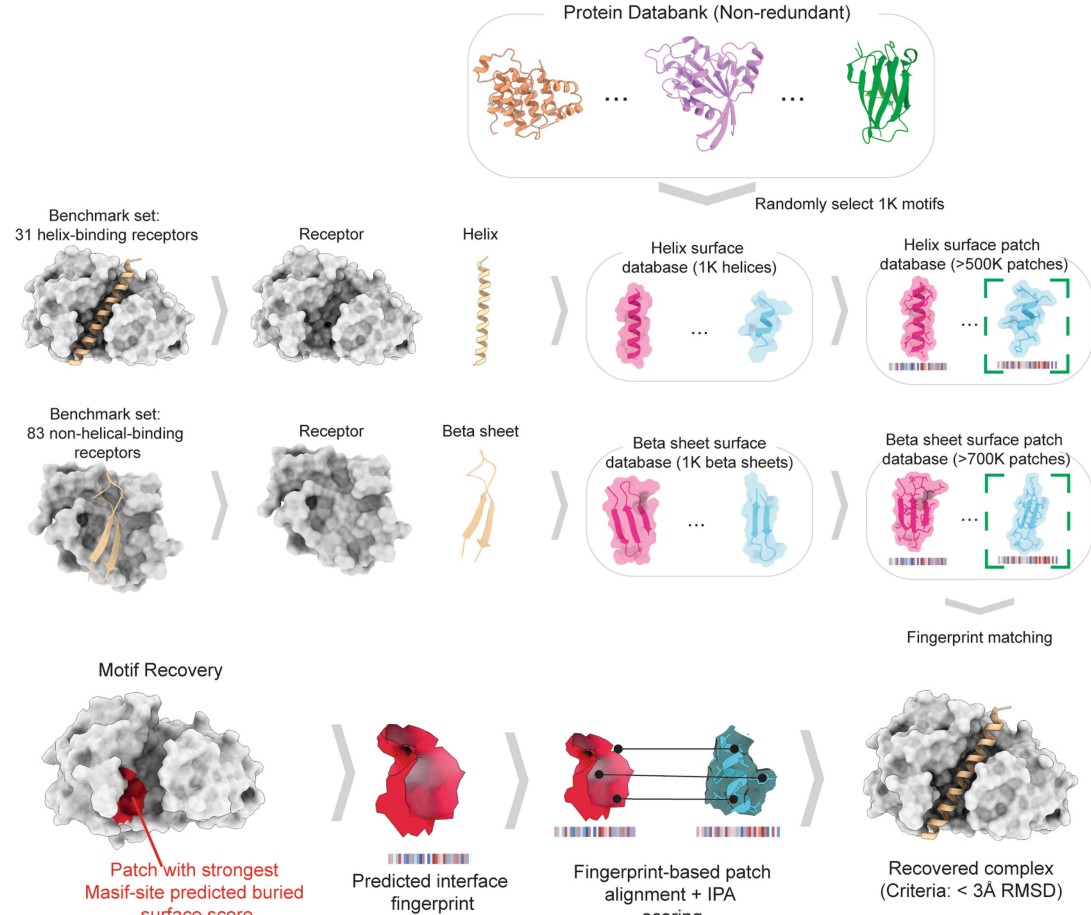

**Extended Data Fig. 2 | MaSIF-seed benchmarking for the discrimination of helical or non-helical binding motifs.** A non-redundant set of 31 helical and 83 non-helical fragments that bind to known protein receptors was selected as a benchmark set to evaluate MaSIF-seed's capacity to recover true binding motifs from decoys, and to correctly rank them among the top results. To generate the decoy set, a non-redundant set of all protein chains in the Protein Data Bank was decomposed into continuous helical segments (left) and two/ three-stranded beta sheets (right), resulting in over 250K helical and over 380K beta motifs, respectively. One thousand of these motifs each were randomly selected to act as decoys in the respective benchmarks. The surfaces for the two sets of 1000 motifs were computed and decomposed into radial patches and for each patch a fingerprint was computed. Recovered complexes were considered correct if an iRMSD < 3 Å was obtained. A comparable procedure was applied to the benchmark tools.

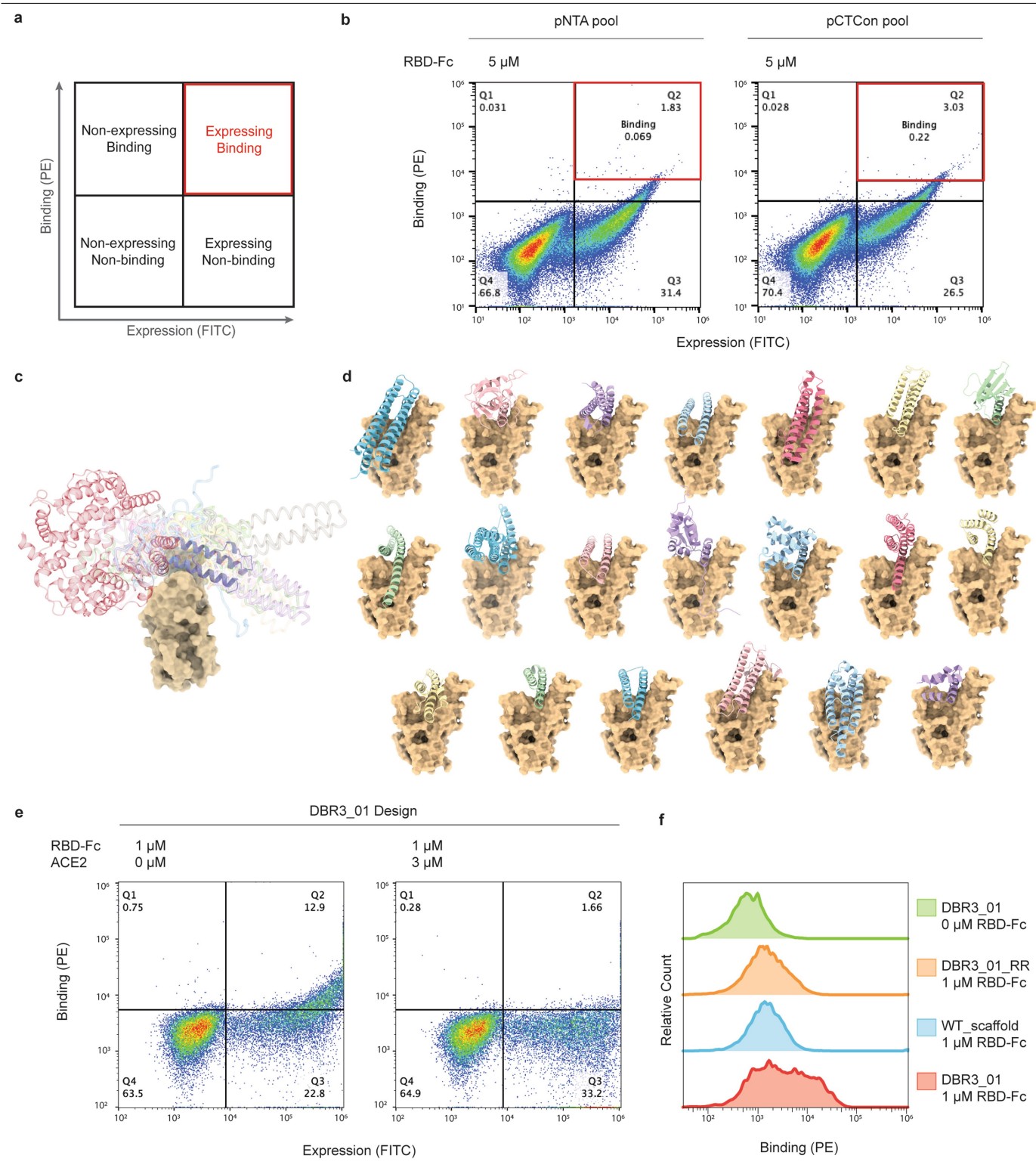

**Extended Data Fig. 3 | RBD-binder designs displayed on yeast. a**, The yeast display protocol utilizes PE to label binding and FITC to label expression. Yeast appearing in the double positive quadrant are considered potential binders and sorted for enrichment. **b**, Pools of approximately 30 designs were displayed on the surface of yeast and the highest binding populations (red box) sorted for further analysis. **c**, Schematic of RBD (wheat) bound to the various members of the library (transparent silhouettes and purple for DBR3_01) and

ACE2 (red) overlapping with the designed binders. **d**, Individual designs DBR1-DBR20. **e**, DBR3_01 design displayed on yeast binds to RBD-Fc (left panel) but the binding is blocked when the RBD-Fc is preincubated with an excess of ACE2, indicating a competitive binding mode. **f**, A point mutant in the binding interface (DBR3_01_RR) and the original scaffold protein (WT_scaffold) show lower binding signal than DBR3_01 with 1 μM RBD-Fc, indicating that the design is engaging the RBD with the predicted interface.

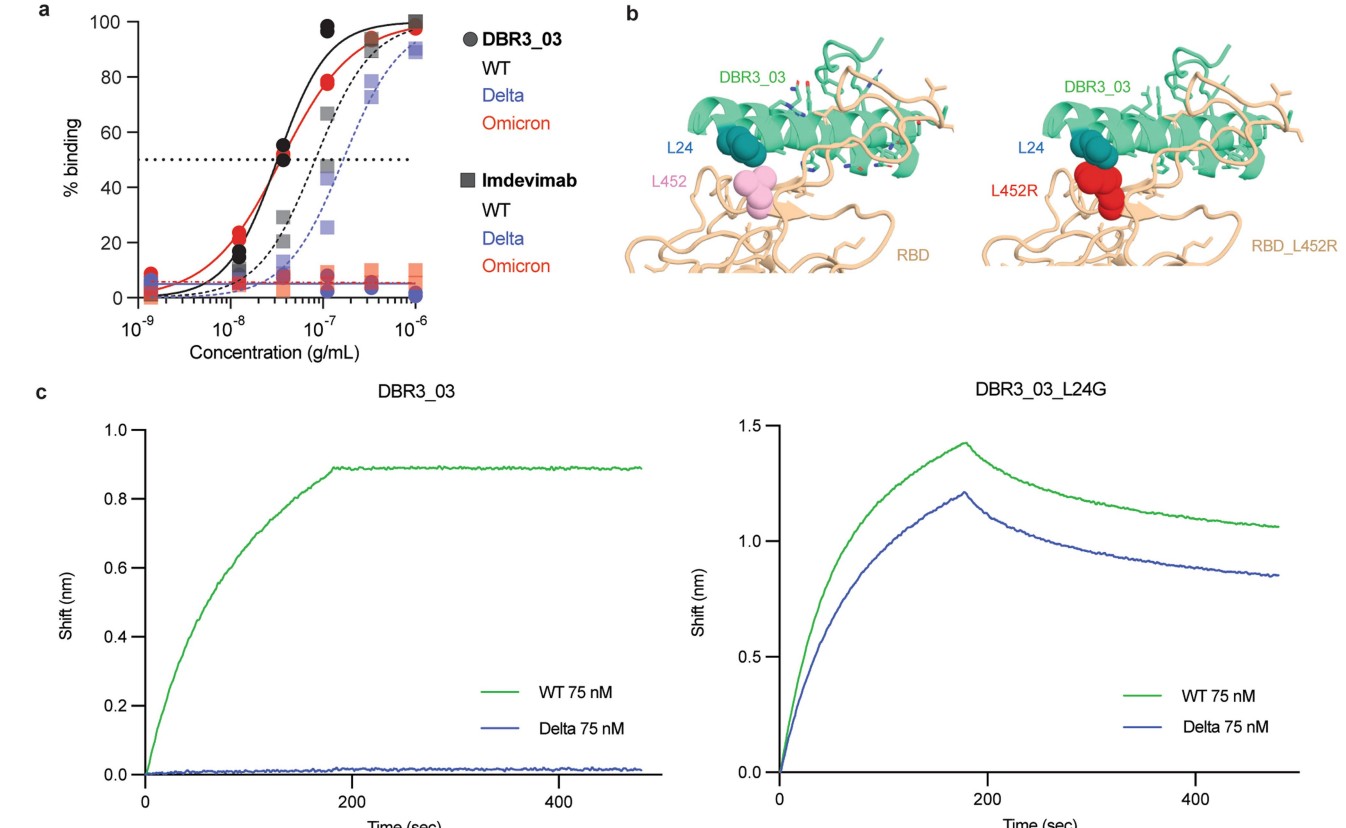

**Extended Data Fig. 4 | DBR3_03 binding is sensitive to the L452R mutation in the spike protein. a**, Luminex binding assay of DBR3_03 or Imdevimab (REGN10987) with beads functionalized with SARS-CoV-2 spike protein of indicated variants. DBR3_03 has an $EC_{50}$ of $3.2e^{-8}$ g ml$^{-1}$ with WT and $3.5e^{-8}$ g ml$^{-1}$ with Omicron. Imdevimab has an $EC_{50}$ of $8.2e^{-8}$ g ml$^{-1}$ with WT and $1.7e^{-7}$ g ml$^{-1}$ with delta. The fits were calculated from technical replicates (n = 2) using a nonlinear four parameter curve fitting analysis. **b**, The L452R mutation on the spike protein leads to a clash with the DBR3_03 binding. A L24G mutation is proposed to avoid the clash. **c**, BLI data with DBR3_03 (WT $K_D$ <0.1 nM, delta $K_D$ not detected) or DBR3_03_L24G (delta $K_D$ = 6 nM, WT $K_D$ = 6 nM) immobilized on the tips, dipped into spike protein of different variants.

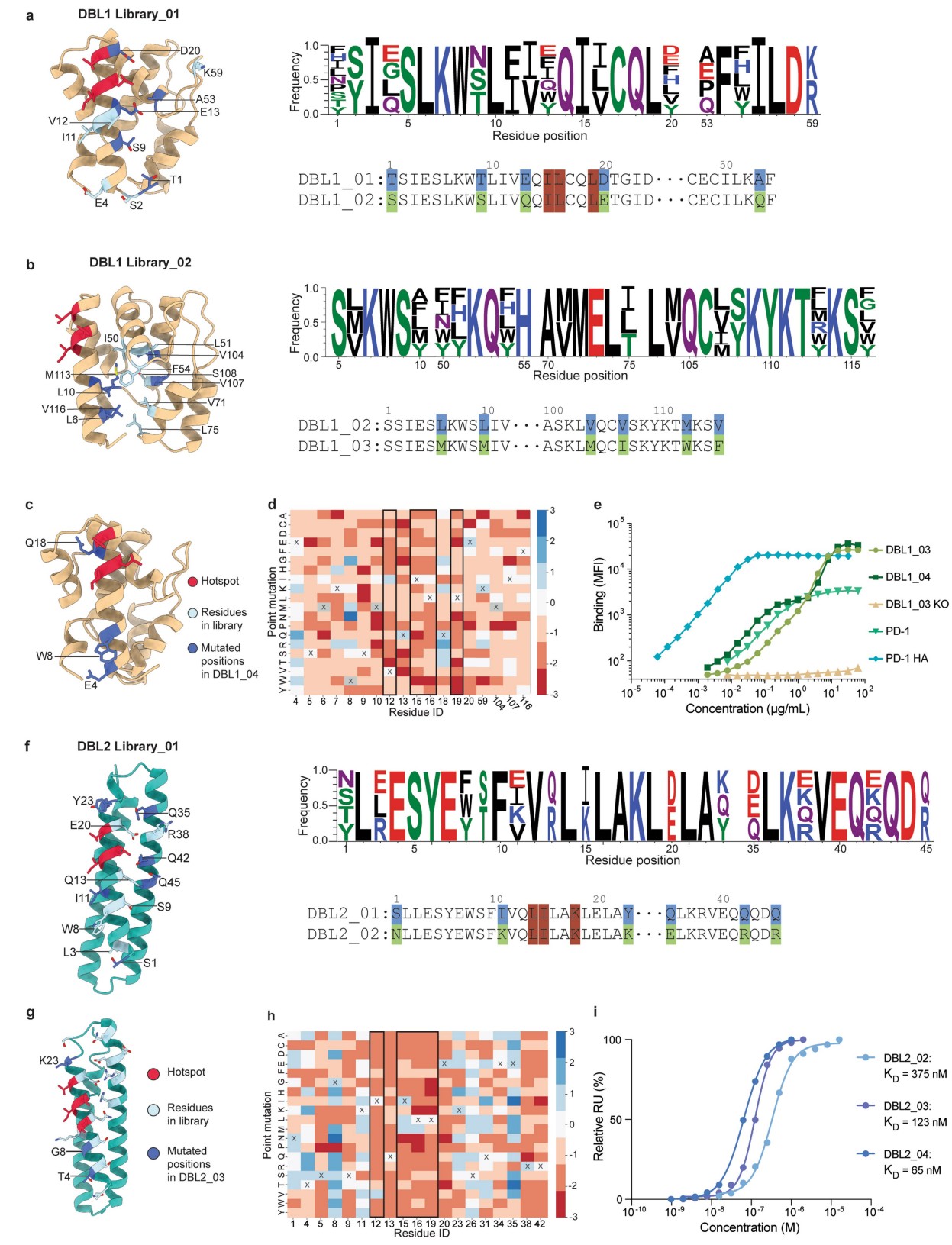

**Extended Data Fig. 5** | See next page for caption.

**Extended Data Fig. 5 | Yeast libraries, SSM and binding data of DBL1/ DBL2_02. a**, Position of targeted residues in the structure of DBL1_01 to improve binding affinity. Logo plot of the allowed mutations in the library and alignment of initial design with library enriched design. **b**, Position of targeted residues in the structure of DBL1_02 to improve core packing. Logo plot of the allowed mutations in the library and alignment of DBL1_02 with library enriched design. **c**, Structural representation of all positions sampled in the SSM library (light blue). The four hotspot residues (red) were also sampled. Three positions were mutated in DBL1_04 (dark blue). **d**, Outcome of the entire SSM library of DBL1_03. Blue indicates enrichment in the binding population, while red shows enrichment in the non-binding population. **e**, Binding of DBL1_03 and DBL1_04 to KARPAS299 cells expressing PD-L1 compared to binding of WT PD-1, a high affinity version of PD-1 (PD-1_HA)[35] and a V12R mutation of DBL1_03 (KO). All proteins contained a Fc domain. **f**, Position of targeted residues in the structure of DBL2_01 to improve binding affinity and solubility. Logo plot of the allowed mutations in the library and alignment of initial design with library enriched design. Hotspot residues red, targeted residues light blue, mutated residues dark blue. **g**, Structural representation of all positions sampled in the SSM library (light blue). The four hotspot residues (red) were also sampled. Three positions were mutated in DBL2_04 (dark blue). Position 35 was not mutated in DBL_04, because all mutations in this position led to the inability of the soluble expression of the protein. **h**, Outcome of the entire SSM library of DBL2_03. Blue indicates enrichment in the binding population, while red shows enrichment in the non-binding population. **i**, Binding affinities measured by SPR for the different versions of DBL2.

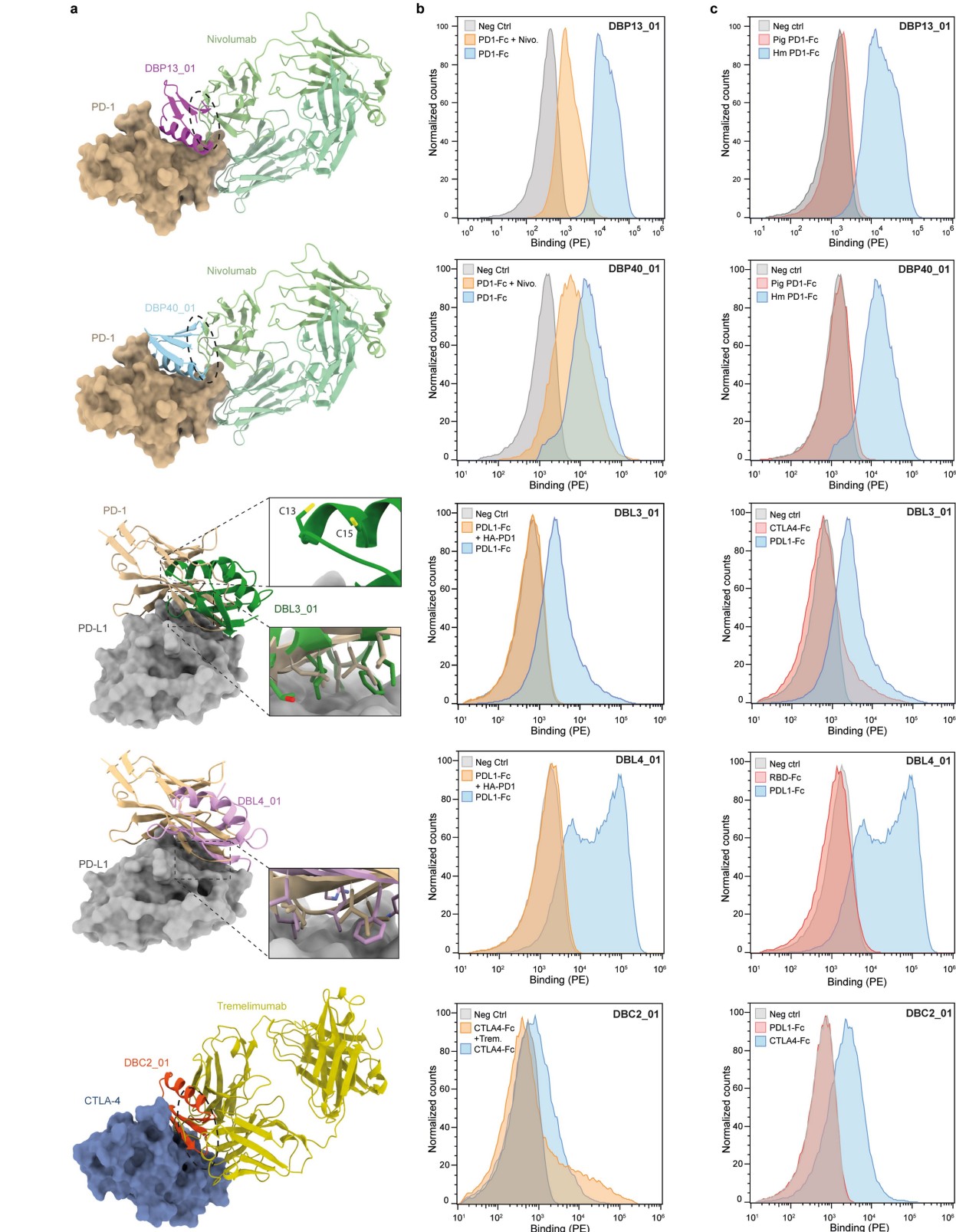

**Extended Data Fig. 6 | Competition and specificity binding assay of the different optimized binders on the surface of yeast. a**, Competition between designed binders and a known protein binder (native binder or monoclonal Fab) in complex with the target structure. **b**, Flow cytometry histograms showing fluorescence signals on the surface of yeast displaying the different binders. Yeasts were labelled with 500 nM or their respective ligand (blue), 500 nM of blocked ligand pre-incubated with 10-fold molar excess of Fab or high-affinity PD-1 (HA-PD-1) (orange) or labelled with secondary antibodies only (grey, Neg Ctrl). **c**, Flow cytometry histograms showing fluorescence signal on the surface of yeast displaying the different binders and labelled with 500 nM of unrelated protein ligand (red) or labelled with secondary antibodies only (grey, Neg Ctrl).

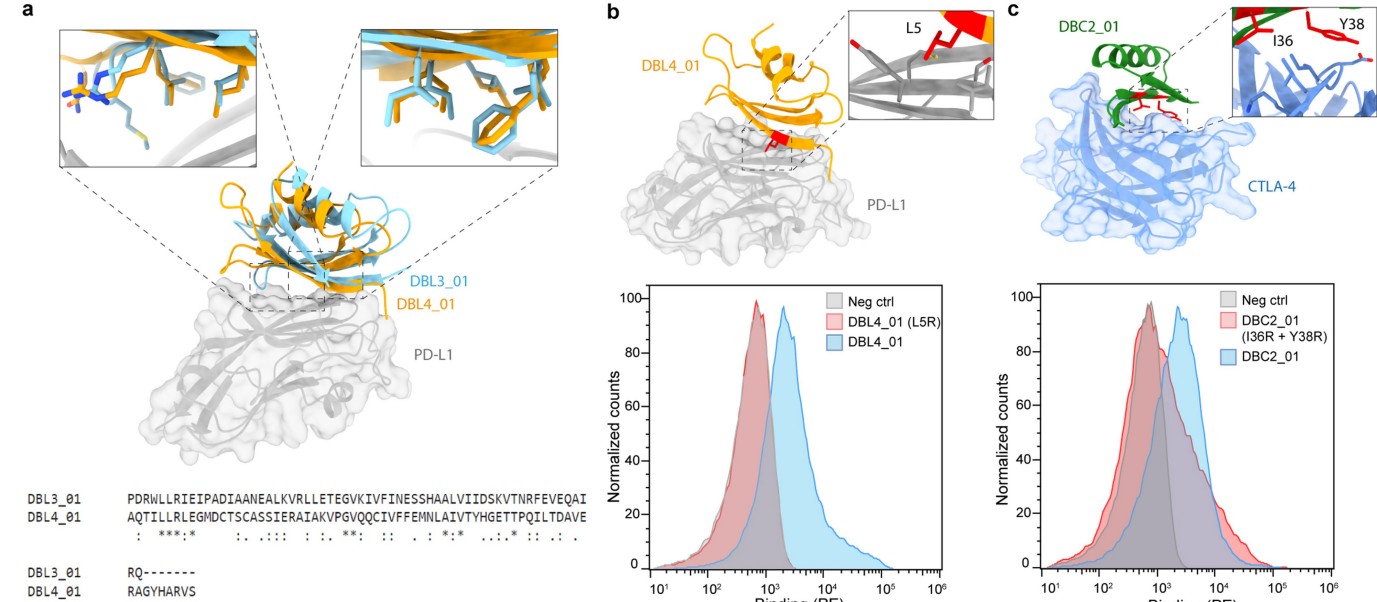

```
DBL3_01    PDRWLLRIEIPADIAANEALKVRLLETEGVKIVFINESSHAALVIIDSKVTNRFEVEQAI
DBL4_01    AQTILLRLEGMDCTSCASSIERAIAKVPGVQQCIVFFEMNLAIVTYHGETTPQILTDAVE
            :  ***:*    :.  .::: :  :.  **:  ::  . : *:* ..:.* :: .:  .

DBL3_01    RQ-------
DBL4_01    RAGYHARVS
           *
```

**Extended Data Fig. 7 | DBL3_01 and DBL4_01 comparison and DBL4_01 and DBC2_01 knock-out mutants. a**, Superposition between DBL3_01 (cyan) and DBL4_01 (orange) in complex with PD-L1 (grey). Multiple sequence alignment of the two designs is shown at the bottom. **b**, DBL4_01 (orange) in complex with PD-L1 (grey) with knock-out mutant highlighted in red. Flow cytometry histograms showing fluorescence signals on the surface of yeast displaying DBL4_01 or the knock-out mutant, compared to unlabelled yeast (Neg Ctrl). **c**, DBC2_01 (green) in complex with CTLA-4 (blue) with two knock-out mutants highlighted in red. Flow cytometry histograms showing fluorescence signals on the surface of yeast displaying DBC2_01 or the knock-out mutants, compared to unlabelled yeast (Neg Ctrl).

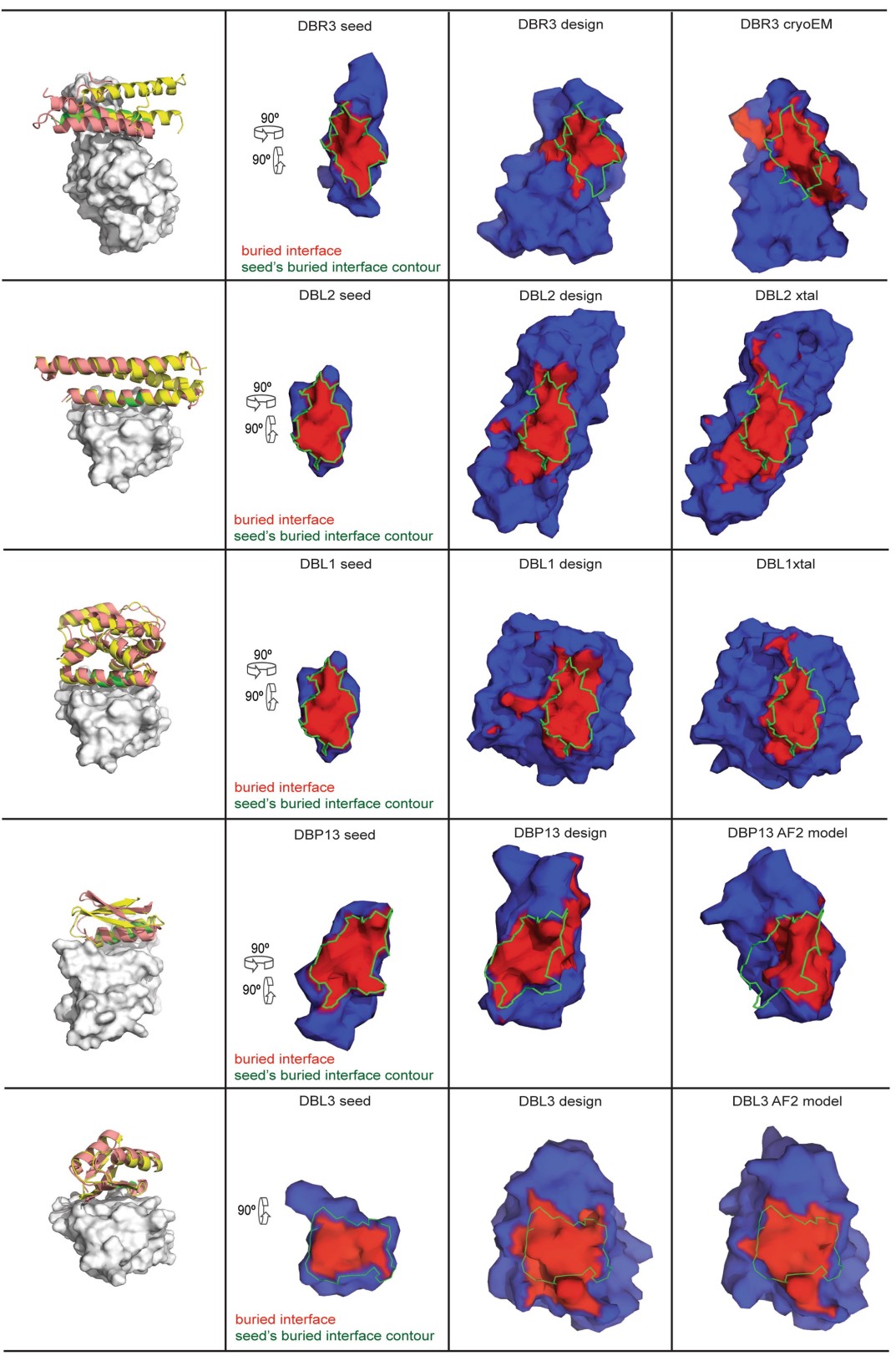

**Extended Data Fig. 8 | Surface comparison between seeds, designs and final/predicted structures.** Buried interfaces of models/structures when in complex with their target are coloured in red, while non-buried regions coloured in blue. The contour of the buried interface of the initial binding seed is drawn in green and is shown for the initial seed, for the designs and for the final/predicted structures.

**Extended Data Table 1 | Crystallographic data collection and refinement statistics**

| | DBL1_03-PD-L1 | DBL2_02-PD-L1 |
|---|---|---|
| **Data collection** | | |
| Space group | P $4_2$ $2_1$ 2 | P $2_1$ $2_1$ $2_1$ |
| Cell dimensions | | |
| $a$, $b$, $c$ (Å) | 97.93, 97.93, 106.11 | 85.41, 116.08, 149.61 |
| $\alpha$, $\beta$, $\gamma$ (°) | 90.00, 90.00, 90.00 | 90.00, 90.00, 90.00 |
| Wavelength (Å) | 0.97889 | 0.97918 |
| Resolution (Å) | 48.97 - 2.85 (2.95 - 2.85) | 41.06 - 3.00 (3.11 - 3.00) |
| Unique reflections | 12591 (1241) | 30347 (2986) |
| $R_{merge}$ | 0.141 (3.126) | 0.165 (2.911) |
| $I / \sigma I$ | 20.7 (1.1) | 12.8 (1.1) |
| Completeness (%) | 99.9 (100.0) | 99.4 (99.7) |
| Redundancy | 25.4 (26.9) | 13.0 (13.1) |
| CC1/2 | 0.998 (0.554) | 0.999 (0.436) |
| | | |
| **Refinement** | | |
| Resolution (Å) | 48.97 - 2.85 | 41.06 - 3.00 |
| No. reflections | 12582 | 30282 |
| $R_{work}$ / $R_{free}$ | 0.3005/0.3220 | 0.2671/0.2945 |
| No. atoms | | |
| Protein | 2619 | 9316 |
| Ligand/ion | 0 | 0 |
| Water | 1 | 0 |
| $B$-factors | | |
| Protein | 124.1 | 134.1 |
| Ligand/ion | - | - |
| Water | 83.5 | |
| R.m.s. deviations | | |
| Bond lengths (Å) | 0.010 | 0.004 |
| Bond angles (°) | 1.250 | 0.700 |
| Ramachandran plot | | |
| Favored (%) | 93.77 | 96.07 |
| Allowed (%) | 6.23 | 3.93 |
| Outliers (%) | 0.00 | 0.00 |

**Extended Data Table 2 | Cryo-EM data collection and model validation statistics**

| | D614G-binder (EMDB-14947) (PDB 7ZSS) | Omicron-binder full (EMDB-14922) (PDB 7ZRV) | Omicron-binder local (EMDB-14930) (PDB 7ZSD) |
|---|---|---|---|
| **Data collection and processing** | | | |
| Magnification | 195K | 165K | 165K |
| Voltage (kV) | 300 | 300 | 300 |
| Electron exposure (e–/Å$^2$) | 80 | 60 | 60 |
| Dose rate (e-/px/s) | 4.53 | 5.4 | 5.4 |
| Exposure times (seconds) | 2.82 | 5.85 | 5.85 |
| Defocus range (µm) | 0.8 – 2.0 | 0.8 – 2.5 | 0.8 – 2.5 |
| Pixel size (Å) | 0.40 | 0.726 | 0.726 |
| Symmetry imposed | C1 | C1 | C1 |
| Micrographs | 20 794 | 22 266 | 22 266 |
| Initial particle images (no.) | 832 816 | 1 820 333 | 1 820 333 |
| Final particle images (no.) | 67 432 | 50 758 | 50 758 |
| Map resolution (Å) | 2.63 | 2.80 | 3.29 |
| FSC threshold | 0.143 | 0.143 | 0.143 |
| Map resolution range (Å) | 30-2.8 | 30-2.4 | 30-2.9 |
| | | | |
| **Refinement** | | | |
| Initial model used (PDB code) | 7BNO | 7QO7 | - |
| Model resolution (Å) | 3.2 | 3.2 | 3.4 |
| FSC threshold | 0.143 | 0.143 | 0.143 |
| Model resolution range (Å) | 30-3.2 | 30-3.2 | 30-3.4 |
| Map sharpening $B$ factor (Å$^2$) | -33.7 | -33.8 | 52.7 |
| Model composition | | | |
| Non-hydrogen atoms | 25948 | 28058 | 2121 |
| Protein residues | 3236/0 | 3429/0 | 261/0 |
| Ligands | NAG:46 | BMA:12 + NAG:76 | NAG:2 |
| $B$ factors (Å$^2$) | | | |
| Protein | 2.00/198.38/88.48 | 0.11/126.79/59.39 | 33.55/111.45/61.63 |
| Ligand | 31.30/175.52/79.67 | 28.80/129.22/79.33 | 58.45/64.51/61.48 |
| R.m.s. deviations | | | |
| Bond lengths (Å) | 0.004(0) | 0.002(3) | 0.003(0) |
| Bond angles (°) | 0.687(35) | 0.534 (18) | 0.577(0) |
| Validation | | | |
| MolProbity score | 1.73 | 1.85 | 1.96 |
| Clashscore | 6.36 | 9.53 | 9.82 |
| Poor rotamers (%) | 0.00 | 0.00 | 0.00 |
| Ramachandran plot | | | |
| Favored (%) | 94.36 | 95.05 | 93.00 |
| Allowed (%) | 5.45 | 4.77 | 7.00 |
| Disallowed (%) | 0.19 | 0.18 | 0.00 |

# Reporting Summary

## Statistics

For all statistical analyses, confirm that the following items are present in the figure legend, table legend, main text, or Methods section.

| n/a | Confirmed | |
|---|---|---|
| ☐ | ☒ | The exact sample size (*n*) for each experimental group/condition, given as a discrete number and unit of measurement |
| ☐ | ☒ | A statement on whether measurements were taken from distinct samples or whether the same sample was measured repeatedly |
| ☒ | ☐ | The statistical test(s) used AND whether they are one- or two-sided<br>*Only common tests should be described solely by name; describe more complex techniques in the Methods section.* |
| ☒ | ☐ | A description of all covariates tested |
| ☒ | ☐ | A description of any assumptions or corrections, such as tests of normality and adjustment for multiple comparisons |
| ☐ | ☒ | A full description of the statistical parameters including central tendency (e.g. means) or other basic estimates (e.g. regression coefficient) AND variation (e.g. standard deviation) or associated estimates of uncertainty (e.g. confidence intervals) |
| ☒ | ☐ | For null hypothesis testing, the test statistic (e.g. *F*, *t*, *r*) with confidence intervals, effect sizes, degrees of freedom and *P* value noted<br>*Give P values as exact values whenever suitable.* |
| ☒ | ☐ | For Bayesian analysis, information on the choice of priors and Markov chain Monte Carlo settings |
| ☒ | ☐ | For hierarchical and complex designs, identification of the appropriate level for tests and full reporting of outcomes |
| ☒ | ☐ | Estimates of effect sizes (e.g. Cohen's *d*, Pearson's *r*), indicating how they were calculated |

*Our web collection on statistics for biologists contains articles on many of the points above.*

## Software and code

Policy information about availability of computer code

| Data collection | XDS v Jan 10, 2022 (BUILT=20220220) package for processing of crystallographic data; Automation program EPU (ThermoFischer Sci., v2.12.1) for cryoEM data collection; Biacore 8K control software (Cytiva, v4.0.8.19879) for measuring surface plasmon resonance; LE-SH800SZFCPL Cell Sorter software (Sony, v2.1.5) for sorting and collecting FACS data; Kaluza for Galios (Beckman Coulter, v1.1.20388.18228) for collecting flowcytometry data; Chromeleon (ThermoFischer Sci, v7.2.10) and Astra (Wyatt Tech., v8.0.2.5) for performing and collecting data of SEC-MALS; GatorOne (Gator Bio, v.7.3.0728) for biolayer interferometry; Python (v3.6.5) as well as TensorFlow (v1.12) and PyMESH (v0.2.1) to run our MaSIF-seed pipeline; Custom scripts to collect protein modeling data are available on Github (https://github.com/LPDI-EPFL/masif_seed) |
|---|---|
| Data analysis | Coot (v0.9.5) for structure building; Prism (GraphPad, v9) for graphs generation; FlowJo (BD Bioscience, v10.8.1) for flowcytometry analysis; PyMol (Schrödinger, v2.0) and ChimeraX (UCSF, v1.3) for protein vizualization and structural graphic generation; MolProbity (v4.5.1) for structure evaluation; Phaser MR and Phenix.refine in Phenix (v1.19.2-4158 and v1.20.1-4487) for crystal structure determination and refinement; cryoSPARC (v3.3.1) for cryoEM structure determination, Biacore Insight Evaluation Software (Cytiva, v4.0.8.19879) for evaluating surface plasmon resonance measurements, Rosetta modelling suite (v3.13) for protein design and analysis; Custom scripts to analyze protein modeling data are available on Github (https://github.com/LPDI-EPFL/masif_seed) |

For manuscripts utilizing custom algorithms or software that are central to the research but not yet described in published literature, software must be made available to editors and reviewers. We strongly encourage code deposition in a community repository (e.g. GitHub). See the Nature Portfolio guidelines for submitting code & software for further information.

## Data

Policy information about availability of data

 All manuscripts must include a data availability statement. This statement should provide the following information, where applicable:

- Accession codes, unique identifiers, or web links for publicly available datasets
- A description of any restrictions on data availability
- For clinical datasets or third party data, please ensure that the statement adheres to our policy

Cryo-EM maps were deposited in the Electron Microscopy Data Bank under the access codes of EMD-14947 (spikeD614G-binder full and spikeD614G-binder local maps), EMD-14922 (spikeOmicron-binder full), and EMD-14930 (spikeOmicron-binder local). Atomic models were deposited in Protein Data Bank under the access codes of PDB-7ZSS (spikeD614G-binder), PDB-7ZRV (spikeOmicron-binder full) and PDB-7ZSD (spikeOmicron-binder local). Crystal structures have been deposited in the Protein Data Bank under accession codes 7XYQ (DBL1_03/PD-L1 complex) and 7XAD (DBL2_02/PD-L1 complex). The PDBbind database (2018 released), PRISM database, ZDock benchmark and SabDab database are available with the following links respectively: http://pdbbind.org.cn/index.php, http://cosbi.ku.edu.tr/prism, https://zlab.umassmed.edu/benchmark/ and http://opig.stats.ox.ac.uk/webapps/sabdab. MaSIF-seed and the Rosetta design scripts are available at https://github.com/LPDI-EPFL/masif_seed. The scaffold database used for grafting the seeds provided by MaSIF-seed is available at https://zenodo.org/record/7643697#.Y-z533ZKhaQ

## Human research participants

Policy information about studies involving human research participants and Sex and Gender in Research.

| Reporting on sex and gender | N/A |
| --- | --- |
| Population characteristics | N/A |
| Recruitment | N/A |
| Ethics oversight | N/A |

Note that full information on the approval of the study protocol must also be provided in the manuscript.

# Field-specific reporting

Please select the one below that is the best fit for your research. If you are not sure, read the appropriate sections before making your selection.

☒ Life sciences        ☐ Behavioural & social sciences        ☐ Ecological, evolutionary & environmental sciences

For a reference copy of the document with all sections, see nature.com/documents/nr-reporting-summary-flat.pdf

# Life sciences study design

All studies must disclose on these points even when the disclosure is negative.

| Sample size | For the protein designs, 16 individual designs were tested for PD-L1, 20 individual designs for SARS-CoV2-RBD and 1500-2000 designs for the one-shot design approach targeting PD-L1, PD-1 and CTLA-4 |
| --- | --- |
| Data exclusions | There is no data exclusion in this study. |
| Replication | Each binding candidate detected by yeast display and/or deep-sequencing showed reproducible results and were also tested with negative controls (unrelated protein ligand or unlabeled yeast) and point mutants. Result reproducibility was also confirmed with alternative methods (e.g. SPR) |
| Randomization | Oligos encoding the one-shot designs were pooled together (one oligopool per target) and then sorted by yeast display |
| Blinding | Researchers were not blinded as this does not apply to our study (no bias expected from cell sorting) |

# Reporting for specific materials, systems and methods

We require information from authors about some types of materials, experimental systems and methods used in many studies. Here, indicate whether each material, system or method listed is relevant to your study. If you are not sure if a list item applies to your research, read the appropriate section before selecting a response.

## Materials & experimental systems

| n/a | Involved in the study |
|---|---|
| ☐ | ☒ Antibodies |
| ☐ | ☒ Eukaryotic cell lines |
| ☒ | ☐ Palaeontology and archaeology |
| ☒ | ☐ Animals and other organisms |
| ☒ | ☐ Clinical data |
| ☒ | ☐ Dual use research of concern |

## Methods

| n/a | Involved in the study |
|---|---|
| ☒ | ☐ ChIP-seq |
| ☐ | ☒ Flow cytometry |
| ☒ | ☐ MRI-based neuroimaging |

# Antibodies

| Antibodies used | 1) Anti-HA, FITC - Ref : A190-138F - Manufacturer : Bethyl - Clone : Unknown<br>2) Anti-V5 mouse - Ref : MA5-15253 - Manufacturer : Invitrogen - Cline : E10/V4RR<br>3) Anti-mouse, FITC - Ref : F0257 - Manufacturer : Sigma - Clone : Polyclonal<br>4) Anti-His, PE - Ref : 130-120-787 - Manufacturer : Miltenyi Biotec - Clone : GG11-8F3.5.1<br>5) Anti-Myc, FITC - Ref : SAB4700448 - Manufacturer : Sigma - Clone : 9E10<br>6) Anti-human IgG, PE - Ref : 12-4998-82 - Manufacturer : Invitrogen - Clone : Polyclonal<br>7) Anti-Human IgG, R-PE - Ref : 109-117-008 - Manufacturer : Jackson ImmunoResearch - Clone : Polyclonal |
|---|---|
| Validation | For the commercially available antibodies, no validation reports were provided other than publications citing the products or examples on manufacturer's webpage :<br>1) 10.21037/atm.2020.03.74<br>2) 10.1038/s41467-021-22969-5<br>3) 10.1101/gad.1575307<br>4) https://www.miltenyibiotec.com/US-en/products/his-antibody-gg11-8f3-5-1.html#pe:100-tests-in-200-ul<br>5) 10.1039/c8nr03970d<br>6) https://www.thermofisher.com/antibody/product/Goat-anti-Human-IgG-Fc-Secondary-Antibody-Polyclonal/12-4998-82<br>7) 10.1038/s41467-020-19231-9 |

# Eukaryotic cell lines

Policy information about cell lines and Sex and Gender in Research

| Cell line source(s) | Karpas-299 purchased from Sigma (Ref: 06072604-1VL) with ECACC (European Collection of Authenticated Cell Cultures) approval<br>Expi293 purchased from ThermoFischer Sci. (Ref: A14635) |
|---|---|
| Authentication | Authenticated by the provider and no additional authentication has been done |
| Mycoplasma contamination | Karpas-299 : Cells were tested negative for mycoplasma contamination by the manufacturer (PCR) and no additional test was performed as the cells were directly used for a single experiment after purchase.<br>Expi299: Cells were tested negative for mycoplasma contamination by the manufacturer (qPCR) and no additional test was performed as the cells were used for protein expression. |
| Commonly misidentified lines<br>(See ICLAC register) | No commonly misidentified lines were used in this study |

# Flow Cytometry

## Plots

Confirm that:

☒ The axis labels state the marker and fluorochrome used (e.g. CD4-FITC).

☒ The axis scales are clearly visible. Include numbers along axes only for bottom left plot of group (a 'group' is an analysis of identical markers).

☒ All plots are contour plots with outliers or pseudocolor plots.

☒ A numerical value for number of cells or percentage (with statistics) is provided.

## Methodology

| Sample preparation | Yeasts cells (EBY-100) were labeled with the protein target and then with an anti-Myc (FITC), anti-V5 (FITC) or anti-HA (FITC) for the display signal and an anti-Fc (PE) or anti-His (PE) for binding to the protein target. Cells were washed with PBS supplemented with 0.1% BSA. See methods for more details |
|---|---|
| Instrument | Sony SH800 (Sorting) and Beckman Coulter Gallios (Analysis) |

| Software | Sony LE-SH800SZFCPL Cell Sorter software (v2.1.5), Kaluza for Gallios (v1.1.20388.18228) and FlowJo (v10.8.1) |
|---|---|
| Cell population abundance | Transformed yeasts underwent several round of sorts for target binding and amplifications in culture (2 to 3 cycles) before being sequenced for the isolation of single binding clones. Single clone candidates (obtained from a single colony on plate or re-transformation of naive yeasts with pure DNA) were then individually tested with controls. |
| Gating strategy | Yeast cells without target labeling served as a negative control and yeasts showing binding signal above this negative threshold were collected (See Extended Data Fig. 3a-b for an example). |

☒ Tick this box to confirm that a figure exemplifying the gating strategy is provided in the Supplementary Information.

