## [Peer Review File · Nature]

Manuscript Title: De novo design of protein interactions with learned surface fingerprints

Reviewer Comments & Author Rebuttals

Reviewer Reports on the Initial Version:

Referee #1:

This study presents a deep-learning framework for designing protein-protein interactions (PPIs). A neural network was trained on known PPIs to identify geometric and chemical features that discriminate interaction and non-interacting protein pairs. The method has a high success rate of predicting natural partners (based on benchmarking in the PDBBind dataset), and the calculations are significantly faster than contemporary docking approaches. This model was then applied to two systems - the RBD of CoV-2 spike and the PD-1/PD-11 interaction on activated T-cells. Binding seeds from a helical library were identified and grafted onto host scaffolds. Low affinity designs to these targets were generated using this approach and then optimized using directed libraries. Neutralization of spike was observed in live omicron-virus assays and structures of designs binding to PD-L1 demonstrating the success of the approach. Given that the approach generated low-affinity starting points that required further optimization - the authors explored a 'one-shot' approach where a larger starting library was used - in this case stronger binding members were identified. Although structures were not determined, AlphaFold models matched the predicted PPIs within a few angstroms RMSD.

There are limitations to the approach acknowledged in the discussion, including restriction to helical seeds and the lack of consideration of surface remodeling upon binding (also cryptic sites that are hidden by protein dynamics). Nevertheless, this work represents a significant advance over prior PPI design approaches.

In a revision it would be useful to the community to understand why this approach outperforms other methods in terms of accuracy of recovering native binders. Given that the benchmarking set is only 31 structures, it would be useful to understand where masif-seed succeeds and fails. Are the false-positives related to the native sequence/structures in any way? Do masif and zdock + zrank2 correctly identify the same proteins? Understanding this could better elucidate the strengths and limitations of using chemical and geometric information from static structures (if that is the issue).

Maybe I missed it, but why was the initial approach developed only for helical seeds: is the idea to avoid conformational epitopes where binding residues are far in sequence? For difficult to drug sites (flat sites), equivalently flat structures such as sheets and barrels might be better pharmacophores.

Referee #2:

Gainza et al. describe a novel approach for protein-protein interaction design. The manuscript is written clearly, and the experimental work, reporting the design of four binders for three targets of

therapeutic interest, is outstanding. The discussion highlights current limitations. I believe this manuscript would be of broad interest to the community and recommend its publication.

Only a few minor points are summarized below:

General points:

1) The manuscript describes very well the experimental characterization, but I could not find as much information for the computational counterpart. In particular, the main text or methods do not specify the size of the training and evaluation datasets for MaSIF-Search, the neural network's architecture, the number of parameters, or how the model was trained. I understand the main text might be limited in length; this information should then be clearly stated in the Methods section. If some of the work was performed in the previous article [ref 19], I would note so, but the text should still include the training specifications.

2) It was unclear to me until I reviewed the GitHub repository and the Methods section that the method only allows finding helical fingerprints. I do not think this is a problem for publication, but I think the abstract or introduction should at least mention that MaSIF-seed specifically split helical fragments. The first mention occurs in the section 'Targeting a predicted neutralization-sensitive site on SARS-CoV-2 spike protein'. A paragraph in the discussion commenting on why focusing on other secondary structural elements like beta sheets is challenging would also be helpful.

Some minor points:

3) After L95, it would be helpful to add a sentence to remind the reader how MaSIF works.

4) Lines 181-184 explain the scaffolding part. I think the authors should at least mention, perhaps in the discussion, that there are other alternative methods now:
<https://www.science.org/doi/10.1126/science.abn2100>.

5) L200 Why is it 6 mutations instead of 7 (4 first round + 3 second round)?

6) L268: Short explanation of how flatness is quantitatively evaluated would be helpful.

Methods Section:

L475-L481: Possibly a suitable paragraph to insert the training specifications.

L542: Large dataset? How large?

L599: MaSIF-search, MaSIF-site and the IPA score were retrained for this benchmark. Please add more details.

Software:

The GitHub page is well-documented. I cannot comment on how the software works since I did not have docker installed on my current workstation. If the authors want to reach a larger user audience they might need to also provide the package through other channels like conda or pip.

Some general ideas:

I have found the manuscript very inspiring, and the following thoughts came to mind. These are not requirements for this manuscript, only a couple of ideas I thought I could share with the authors.

1) Perhaps the authors, in future work, would benefit from backpropagating this fine-tuned RoseTTAFold model (<https://www.science.org/doi/10.1126/science.abn2100>) as an alternative to Rosetta MotifGraft. Another advantage is that it would allow generating novel scaffolds not included in the training set (1300 monomeric scaffolds in this case).

2) Since the description of the neural networks is not clear in the Methods section, I do not know if the net has generative capabilities. Perhaps the net could directly generate novel fingerprints given a fingerprint partner? This would avoid searching in the 140M database of fingerprints and generate de novo motifs. One issue would be then to translate the fingerprint to an actual structure and I am not sure at the moment how that could be done.

Referee #3:

The field of de novo design of protein-protein interactions is has been limited by low hit rates and the requirement for extensive affinity maturation of successful leads through phage or yeast display libraries. As one example, the recent paper from Cao et al. (2022) required in vitro selection from pools of tens of thousands of designs. So really computational design at this point is most effective at reducing the size of combinatorial libraries needed for screening. Gainza and colleagues have previously reported a deep learning-based method, MaSIF, to generate fingerprints of chemical and geometric features of a protein surface. They had applied this method to predict ligand-interaction sites, to predict protein-protein interaction (PPI) sites, and to predict the structural configurations of protein-protein complexes. In the present work, they extend the method to generation of novel protein-protein interactions. They do so in three computational steps: prediction of sites with high target-binding propensity using their previously reported method; search for complementary structural motifs, using a new method called MaSIF-seed; grafting of seeds onto protein scaffolds using established techniques. This is followed by further in silicon and in vitro protein engineering to yield de novo binders of moderately high affinity (10^{-7} - 10^{-8} M). Binding modes are validated by crystallography and cryoEM.

In the first two attempts described, CoV-2 spike protein and PD-L1, barely measurable binding is achieved following the MaSIF approach and optimization in Rosetta; extensive in vitro engineering yields binders in the range of 80 - 265 nM. Although conceptually similar things have been done, most notably by the Baker lab, this remains an important frontier, and it is impressive to see that the authors have done this using a different computational approach. The authors then adapted their approach for a third target, PD-1, and achieved a 4 μ M binder in a “one-shot” experiment; this is the

most impressive result in the paper. However, it is not clear from this single example whether the “one-shot” approach is generalizable to other protein interfaces, but if so it would be a significant advance for the field. Furthermore there is no structure of the “out of the box” complex, and that would be very interesting to see and validate the design and understand how it could be improved.

Despite the promise of interaction fingerprints as a basis for de novo design, I do not think that this paper has adequately proven that their approach is significantly better than other approaches for “out of the box” protein design. The reliance on helical motifs is a limitation of the method. Further development along these lines should be done for publication in a journal like Nature. I do think that this paper will be of immediate general interest in the fields of protein engineering and protein design.

Other concerns:

The authors claim in the introduction (line 86) that their method “reduces the complexity for hotspot... grafting”. Conceptually, this claim appears to rely on the use of alpha-helical seeds, which the authors acknowledge limits the success of their method. It is also not clear to the reader that, in practice, the success of grafting is greater than would be seen using a rotamer interaction field-based method.

The cryoEM data for the spike protein structures is clearly presented and supports the predicted poses of binders. The crystallographic data for PDL-1 is less clear. Crystal structures are at low resolution and have high refinement R factors. At this resolution, topology errors in the structure can occur through misalignment of residues and missing loops between helices. A supplementary figure showing electron density maps that support the binding modes depicted in Fig. 3c should be provided. Authors should report CC1/2 for the high resolution bin of each crystal structure. The high I/sigma of the DBL2_02-PD-L1 structure suggests that higher resolution data should be included.

Minor concern:

Some references in the first paragraph are duplicated, including preprint and final copies of the same work.

Author Rebuttals to Initial Comments:

Referees' comments and responses

Referee #1:

1.1: This study presents a deep-learning framework for designing protein-protein interactions (PPIs). A neural network was trained on known PPIs to identify geometric and chemical features that discriminate interaction and non-interacting protein pairs. The method has a high success rate of predicting natural partners (based on benchmarking in the PDBind dataset), and the calculations are significantly faster than contemporary docking approaches. This model was then applied to two systems - the RBD of CoV-2 spike and the PD-1/PD-11 interaction on activated T-cells. Binding seeds from a helical library were identified and grafted onto host scaffolds. Low affinity designs to these targets were generated using this approach and then optimized using directed libraries. Neutralization of spike was observed in live omicron-virus assays and structures of designs binding to PD-L1 demonstrating the success of the approach. Given that the approach generated low-affinity starting points that required further optimization - the authors explored a 'one-shot' approach where a larger starting library was used - in this case stronger binding members were identified. Although structures were not determined, AlphaFold models matched the predicted PPIs within a few angstroms RMSD.

There are limitations to the approach acknowledged in the discussion, including restriction to helical seeds and the lack of consideration of surface remodeling upon binding (also cryptic sites that are hidden by protein dynamics). Nevertheless, this work represents a significant advance over prior PPI design approaches.

R: We thank the reviewer for the positive assessment of our work.

1.2: In a revision it would be useful to the community to understand why this approach outperforms other methods in terms of accuracy of recovering native binders. Given that the benchmarking set is only 31 structures, it would be useful to understand where masif-seed succeeds and fails. Are the false-positives related to the native sequence/structures in any way? Do masif and zdock + zrank2 correctly identify the same proteins? Understanding this could better elucidate the strengths and limitations of using chemical and geometric information from static structures (if that is the issue).

R: MaSIF-seed was designed on the premise of 'idealized' protein-protein interfaces, where the majority of the contacts occur at a buried surface 'core' with few waters, and this buried surface core can be captured by a radial geodesic patch with 12 Å radius (roughly 400 Å²). The network is trained to produce similar fingerprints for complementary patches. Thus, if a PPI cannot be largely captured by a radial patch (meaning different discontinuous patches

compose the interface) or if the interface has poor shape complementarity (i.e., heavily solvated), MaSIF will not perform as well. To illustrate this point we performed additional analysis on the results of the benchmark shown in Fig. R1 (added as Supplementary Fig. S5 in the manuscript). In Fig. R1a we plotted all 31 cases from our helical benchmark according to the area of the maximum circumscribed patch inside the buried interface of the molecular surface (y-axis), and the median shape complementarity of all points in that patch (x-axis). In this plot we can see that the emergent trend is that those benchmark cases where MaSIF failed to recover the native binder (red Xs) tend to have either a very poor complementarity or their interface is not primarily circumscribed to a large surface patch.

We also found that ZDock/ZRank2 tends to perform best on those cases where the interface is circumscribed into a large patch with high complementarity (Fig. R1b). Indeed, there is a large overlap between the cases successfully solved by ZDock/ZRank2 and MaSIF-seed (table in Fig. R1c). For example, out of the cases correctly ranked by both MaSIF-seed and ZDock/ZRank2 in the Top 100 results, they overlap in close to 80%.

Overall, we find that our method performs well whenever: I) it is able to identify the protein site; II) the site entails a large fraction of the protein interface; III) the interface has high complementarity. In the two benchmarks presented in our manuscript, the number of decoy complexes was limited to 1000 due to the run times required by the competing methods. However, in experiments we performed where we increased the number of decoys we found that, while MaSIF-seed continued to identify the majority of the correct solutions as top-ranked, the performance of competing methods consistently deteriorated with additional decoys. We have added a summary of our findings in the main manuscript.

Figure R1. Analysis of successful/failed helical benchmark cases and comparison between MaSIF-seed and ZDock/ZRank2 performance. **a-b**, Plotting of Top 1, Top 10, Top 100, and Incorrect (ranked < 100) showing the maximum circumscribed patch area in the buried interface (y-axis) and the shape complementarity of that patch (x-axis) for **a**, MaSIF-seed, and **b**, ZDock/ZRank2. **c**, Comparison of cases solved by only MaSIF-seed, only ZDock/ZRank2, or both MaSIF-seed and ZDock/ZRank2 in the Top 1, top 10 or Top 100 rank. **d**, Analysis of two cases that showed both a large circumscribed patch and high complementarity at that patch where MaSIF-seed failed. In both cases, MaSIF-seed failed because it identified a different site as the top site, but increasing the number of sites

explored to the top two resulted in successful predictions. The white dots on the surface denote the predicted site patches.

1.3: Maybe I missed it, but why was the initial approach developed only for helical seeds: is the idea to avoid conformational epitopes where binding residues are far in sequence? For difficult to drug sites (flat sites), equivalently flat structures such as sheets and barrels might be better pharmacophores.

R: We initially selected alpha helical motifs due to their ease of grafting onto scaffold proteins, which in some sense is related to the point of the reviewer, e. g. residues far in sequence are harder for grafting strategies. However, there is no technical limitation in our pipeline that precludes using any type of structural motif independently of secondary structure content or number of segments of the motif, which is clearly a strength of the approach. The MaSIF pipeline itself is agnostic to backbone conformations, as it only receives the surface as input. Indeed, in the revised manuscript, we benchmarked our method along with ZDock and ZDock/ZRank2 (Updated Table 1 in the main manuscript) on a much larger set of additional 83 receptors that engage non-helical binding motifs. The performances we obtained are on par with those observed for helical motifs and remain superior to other docking methods. In addition, we show that our design pipeline can be extended using non-helical binding seeds by providing several examples of one-shot protein designs confirmed experimentally. This data is presented in the revised manuscript in Fig. 4 and discussed in more detail in point 3.1 in response to reviewer's requests.

Referee #2:

2.1: Gainza et al. describe a novel approach for protein-protein interaction design. The manuscript is written clearly, and the experimental work, reporting the design of four binders for three targets of therapeutic interest, is outstanding. The discussion highlights current limitations. I believe this manuscript would be of broad interest to the community and recommend its publication.

R: We thank the reviewer for the positive assessment of our work.

Only a few minor points are summarized below:

General points:

2.2: The manuscript describes very well the experimental characterization, but I could not find as much information for the computational counterpart. In particular, the main text or

methods do not specify the size of the training and evaluation datasets for MaSIF-Search, the neural network's architecture, the number of parameters, or how the model was trained. I understand the main text might be limited in length; this information should then be clearly stated in the Methods section. If some of the work was performed in the previous article [ref 19], I would note so, but the text should still include the training specifications.

R: We have improved the description of the computational pipeline by including more details in the main text and extending the methods section.

2.3: It was unclear to me until I reviewed the GitHub repository and the Methods section that the method only allows finding helical fingerprints. I do not think this is a problem for publication, but I think the abstract or introduction should at least mention that MaSIF-seed specifically split helical fragments. The first mention occurs in the section 'Targeting a predicted neutralization-sensitive site on SARS-CoV-2 spike protein'. A paragraph in the discussion commenting on why focusing on other secondary structural elements like beta sheets is challenging would also be helpful.

R: We thank the reviewer for this comment, which was also shared by other reviewers. As we mentioned above in point 1.3, the MaSIF-seed pipeline is not limited to helical motifs and is in fact agnostic to backbone conformations. In the initial attempts in our work we used helical motifs to simplify the motif grafting stage, which was not the focus of our work. In this revision we present new computational benchmarks that show that the MaSIF framework can be used with any type of motif. In the computational benchmark we show that the performance with non-helical motif remains high (revised Table 1 and supplementary Table 1). Additionally, we have also designed and experimentally characterized a new set of “one-shot” binders which used non-helical motifs as binding seeds for design, and we obtained several binders against 3 different targets, including a new target, CTLA-4. These new results highlight that helices may in fact not be the most favorable structural space to identify binding motifs.

Some minor points:

2.4: After L95, it would be helpful to add a sentence to remind the reader how MaSIF works.

R: We have added the following sentences to give more detail about the MaSIF framework: “Within this framework we developed the MaSIF-site tool¹⁷ to predict areas with propensity to form protein-protein interactions on the surface of proteins. MaSIF-site receives as input a protein decomposed into patches and outputs a per-vertex regression score on the propensity of each point to become a buried surface area within a PPI. MaSIF-search was designed as a Siamese neural network architecture¹⁸ trained to produce similar fingerprints

for the target patch vs. the binder patch, and dissimilar fingerprints for the target patch vs. the random patch.”

2.5: Lines 181-184 explain the scaffolding part. I think the authors should at least mention, perhaps in the discussion, that there are other alternative methods now: <https://www.science.org/doi/10.1126/science.abn2100>.

R: We have included this reference in the revised discussion.

2.6: L200 Why is it 6 mutations instead of 7 (4 first round + 3 second round?)?

R: This was addressed by clarifying that one mutation in the second round occurred in the same position as the first, therefore only 6 total mutations from the original binder.

2.7: L268: Short explanation of how flatness is quantitatively evaluated would be helpful.

R: The flatness calculation is explained in the Supplementary Figure 20 caption, now referred to in the text. We have also added a new section in the method detailing exactly how this was computed titled “Calculation of surface planarity”, and a new directory in the repository called “scripts” now contains the Jupyter Notebook used for the calculations.

Methods Section:

2.8: L475-L481: Possibly a suitable paragraph to insert the training specifications.

R: The training specifications for both modules (MaSIF-site and MaSIF-search), as well as details for the Interface Post Alignment (IPA) NN, are now described in the updated methods section.

2.9: L542: Large dataset? How large?

R: The number of complexes included in the dataset has now been stated under the section “Training of MaSIF-search:
“MaSIF-search was trained on a dataset of 6001 protein-protein interactions ...”

2.10: L599: MaSIF-search, MaSIF-site and the IPA score were retrained for this benchmark. Please add more details.

A description of the three neural networks was added.

Software:

2.11: The GitHub page is well-documented. I cannot comment on how the software works since I did not have docker installed on my current workstation. If the authors want to reach a larger user audience they might need to also provide the package through other channels like conda or pip.

R: MaSIF was built on top of the PyMesh library, which provides a series of powerful tools for manipulating 3D surfaces. Unfortunately, PyMesh is difficult to install, impossible in our hands through automated install tools like pip, and its authors recommend using it through its docker image. The MaSIF docker image is built directly from the docker image provided by the PyMesh creators. In the future we will invest further efforts in making our package more accessible.

Some general ideas:

I have found the manuscript very inspiring, and the following thoughts came to mind. These are not requirements for this manuscript, only a couple of ideas I thought I could share with the authors.

2.12: Perhaps the authors, in future work, would benefit from backpropagating this fine-tuned RoseTTAFold model (<https://www.science.org/doi/10.1126/science.abn2100>) as an alternative to Rosetta MotifGraft. Another advantage is that it would allow generating novel scaffolds not included in the training set (1300 monomeric scaffolds in this case).

R: We thank the reviewer for this suggestion, and we will consider this option for extending our strategy to generate scaffolds more tailored to the binding motifs which will also open the possibility of employing more complex binding motifs for the de novo design of binders.

2.13: Since the description of the neural networks is not clear in the Methods section, I do not know if the net has generative capabilities. Perhaps the net could directly generate novel fingerprints given a fingerprint partner? This would avoid searching in the 140M database of fingerprints and generate de novo motifs. One issue would be then to translate the fingerprint to an actual structure and I am not sure at the moment how that could be done.

R: This is a very exciting perspective for further developments. Currently, MaSIF does not have generative capabilities, it is however something for which we can train the network and that could provide novel routes for the de novo design of binders.

Referee #3:

The field of de novo design of protein-protein interactions is has been limited by low hit rates and the requirement for extensive affinity maturation of successful leads through phage or yeast display libraries. As one example, the recent paper from Cao et al. (2022) required in vitro selection from pools of tens of thousands of designs. So really computational design at this point is most effective at reducing the size of combinatorial libraries needed for screening. Gainza and colleagues have previously reported a deep learning-based method, MaSIF, to generate fingerprints of chemical and geometric features of a protein surface. They had applied this method to predict ligand-interaction sites, to predict protein-protein interaction (PPI) sites, and to predict the structural configurations of protein-protein complexes. In the present work, they extend the method to generation of novel protein-protein interactions. They do so in three computational steps: prediction of sites with high target-binding propensity using their previously reported method; search for complementary structural motifs, using a new method called MaSIF-seed; grafting of seeds onto protein scaffolds using established techniques. This is followed by further in silicon and in vitro protein engineering to yield de novo binders of moderately high affinity (10⁻⁷-10⁻⁸ M). Binding modes are validated by crystallography and cryoEM.

3.1: In the first two attempts described, CoV-2 spike protein and PD-L1, barely measurable binding is achieved following the MaSIF approach and optimization in Rosetta; extensive in vitro engineering yields binders in the range of 80 - 265 nM. Although conceptually similar things have been done, most notably by the Baker lab, this remains an important frontier, and it is impressive to see that the authors have done this using a different computational approach. The authors then adapted their approach for a third target, PD-1, and achieved a 4 μM binder in a “one-shot” experiment; this is the most impressive result in the paper. However, it is not clear from this single example whether the “one-shot” approach is generalizable to other protein interfaces, but if so it would be a significant advance for the field.

R: We thank the reviewer for the positive feedback about our work. To address the requests of the reviewer, we have now employed our design pipeline using different structural motifs (beta-sheet seeds) and obtained several new “one-shot” binders for two different targets (PD-L1, CTLA4). Notably, we have performed deeper experimental characterization in one of these binders targeting PD-L1, showing that the design engages the target with the expected site tested by mutagenesis and competition experiments. Furthermore, we

determined the apparent binding affinity at the surface of yeast obtaining an apparent K_D of 28 nM, which is 42-fold higher than the reference binder (high affinity version of PD-1) at the surface of yeast. The high affinity PD-1 has a reported affinity of 0.1 nM measure by SPR (Maute et al., PNAS, 2015), implying that our designed binder may reach sub- μ M affinity. The designed binding interaction by MaSIF is also accurately predicted by AF which reinforces the confidence in the designed model. These data are now shown in revised Fig. 4 of the main manuscript. For the remaining designs we performed minimal experimental characterization where we show that the designs engage the expected sites by competition experiments, shown in Supplementary Fig. S28 and S31. We however note that the AF predictions are less confident in the binding mode, which may simply be a demonstration of some of the limitations of AF and more work will be necessary to characterize the design's binding modes. Overall, the new results added to the manuscript are a demonstration of two key aspects: I) robustness of the "one-shot" design pipeline; II) generalizability for the design of binding interfaces with structural motifs that go beyond helical motifs and that technically can be expanded to any other structural segments. Altogether, we reckon that the new body of data presented in the revised manuscript appropriately addresses the reviewer's concerns.

3.2 Furthermore there is no structure of the "out of the box" complex, and that would be very interesting to see and validate the design and understand how it could be improved.

R: We are in full agreement with the reviewer in terms of the need for more structural data to better characterize the binding modes of the de novo designed binders. We have put our best efforts both in x-ray crystallography and cryo-EM (including systems to increase the molecular weight of the complex) to solve a high-resolution structure. Unfortunately, none of our efforts have yielded high-quality data for structural determination. Due to time and resource constraints, currently we present a biochemically-based validation for the new designs, and in the future more efforts for structural characterization will be continued for the structural characterization of these designs.

3.3 Despite the promise of interaction fingerprints as a basis for de novo design, I do not think that this paper has adequately proven that their approach is significantly better than other approaches for "out of the box" protein design. The reliance on helical motifs is a limitation of the method. Further development along these lines should be done for publication in a journal like Nature. I do think that this paper will be of immediate general interest in the fields of protein engineering and protein design.

R: We have now added new experimental and computational data that illustrates the generalizability and the robustness of the MaSIF design strategy we proposed. These points have been discussed in detail throughout the responses to reviewers and also substantiated

with additions to the manuscript. Therefore we hope that the reviewer will agree that we have now achieved the next-level requirements for publication in a journal like Nature.

Other concerns:

3.4 The authors claim in the introduction (line 86) that their method “reduces the complexity for hotspot... grafting”. Conceptually, this claim appears to rely on the use of alpha-helical seeds, which the authors acknowledge limits the success of their method. It is also not clear to the reader that, in practice, the success of grafting is greater than would be seen using a rotamer interaction field-based method.

R: Protein interfaces are formed by both side chain and backbone interactions, yet hotspot grafting-based methods focus primarily on the side chain contribution. In addition, hotspot grafting-based methods present three challenges, namely transplanting the side chains on a backbone, stabilizing these side chain conformations in the transplanted backbone, and ensuring that the transplanted backbone is compatible with the interface and forms tight packing with the target, without leaving cavities in the interface. The MaSIF approach does not pose such limitations as the surface fingerprints have embedded backbones. Our more complete surface-centric perspective ensures that both backbone and side chains form a well-packed interaction with the target, while the focus on seeds from existing structures ensures that side chain conformations are energetically favored and stabilized by their environment. As we have shown in this revision, the surface fingerprints are not limited to helical motifs. In terms of the experimental success rates, hotspot-based approaches have always required extensive in vitro optimization, while with the MaSIF approach we are consistently obtaining binders straight from computational calculations.

3.5: The cryoEM data for the spike protein structures is clearly presented and supports the predicted poses of binders. The crystallographic data for PDL-1 is less clear. Crystal structures are at low resolution and have high refinement R factors. At this resolution, topology errors in the structure can occur through misalignment of residues and missing loops between helices. A supplementary figure showing electron density maps that support the binding modes depicted in Fig. 3c should be provided. Authors should report CC1/2 for the high resolution bin of each crystal structure. The high I/sigma of the DBL2_02-PD-L1 structure suggests that higher resolution data should be included.

R: We thank the reviewer for this input, we have included the electron density maps in Figure S26 to demonstrate that the binding modes of the binders could be unambiguously assigned. We also reprocessed the crystallographic datasets to higher resolution, to conform to present day data standards, using I/sigma values over 1.0 and CC1/2 over 0.3 as cut-offs. The newly refined structure will be deposited in the PDB.

Minor concern:

3.6 Some references in the first paragraph are duplicated, including preprint and final copies of the same work.

R: Thank you for this note, these references have been corrected.

Reviewer Reports on the First Revision:

Referee #1:

The revised manuscript clarifies my questions from the initial draft. Suppl. Fig. 5 is informative in understanding where masif and previous approaches were challenged. Please add a color bar to Suppl. Fig. 5d to explain the surface color values and reference the structures shown.

Referee #2:

The authors have addressed my concerns in a positive manner, including more examples with motifs other than helical partners.

Referee #3:

The authors have satisfactorily addressed my concerns. I think this manuscript is appropriate for publication.

Author Rebuttals to First Revision:

Referees' comments and responses – Second revisions

Referee #1:

The revised manuscript clarifies my questions from the initial draft. Suppl. Fig. 5 is informative in understanding where masif and previous approaches were challenged. Please add a color bar to Suppl. Fig. 5d to explain the surface color values and reference the structures shown.

R: We thank the reviewer for his support. The figure has been corrected, as requested.

Referee #2:

The authors have addressed my concerns in a positive manner, including more examples with motifs other than helical partners.

R: We thank the reviewer for his positive feedback.

Referee #3:

The authors have satisfactorily addressed my concerns. I think this manuscript is appropriate for publication.

R: We thank the reviewer for the positive assessment of our work.